# Recent changes of relative humidity: regional connection with land and ocean processes

Sergio M. Vicente-Serrano[1], Raquel Nieto[2], Luis Gimeno[2], Cesar Azorin-Molina[3], Anita Drumond[2], Ahmed El Kenawy[1,4], Fernando Dominguez-Castro[1], Miquel Tomas-Burguera[5], Marina Peña-Gallardo[1]

*[1] Instituto Pirenaico de Ecología, Consejo Superior de Investigaciones Científicas (IPE–CSIC), Zaragoza, Spain; [2]Environmental Physics Laboratory, Universidade de Vigo, Ourense, Spain. [3]Regional Climate Group, Department of Earth Sciences, University of Gothenburg, Sweden. [4]Department of Geography, Mansoura University, Mansoura, Egypt; [5]Estación Experimental Aula Dei, Consejo Superior de Investigaciones Científicas (EEAD-CSIC), Zaragoza, Spain;*

* Corresponding author: svicen@ipe.csic.es

**Abstract.** We analyzed changes in surface relative humidity (RH) at the global scale from 1979 to 2014 using both observations and ERA-Interim dataset. We compared the variability and trends of RH with those of land evapotranspiration and ocean evaporation in moisture source areas across a range of selected regions worldwide. The sources of moisture for each particular region were identified by integrating different observational data and model outputs into a lagrangian approach. The aim was to account for the possible role of changes in air temperature over land, in comparison to sea surface temperature (SST), but also the role of land evapotranspiration and the ocean evaporation on RH variability. The results demonstrate that the patterns of the observed trends in RH at the global scale cannot be linked to a particular individual physical mechanism. Our results also stress that the different hypotheses that may explain the decrease in RH under a global warming scenario could act together to explain recent RH trends. Albeit with the uncertainty in establishing a direct causality between RH trends and the different empirical moisture sources, we found that the observed decrease in RH in some regions (e.g.) can be linked to lower water supply from land evapotranspiration. In contrast, the empirical relationships also suggest that RH trends in other target regions (e.g.) are mainly explained by the dynamic and thermodynamic mechanisms related to the moisture supply from the oceanic source regions. Overall, while this work gives insights on the connection of RH trends with oceanic and continental processes at the global scale, further investigation to assess the contribution of both dynamic and thermodynamic factors to the evolution of RH over continental regions at more-detailed spatial scales is desired.

**Key-words:** Relative humidity; Evaporation; Evapotranspiration; Moisture; Trends; Oceans.

## 1. Introduction

Relative Humidity (RH) is a key meteorological parameter that determines the aerodynamic component of the atmospheric evaporative demand (AED) (Wang and Dickinson, 2012; McVicar et al., 2012a). As such, changes in RH may impact significantly the evolution of the AED (Vicente-Serrano et al., 2014a), with particular implications for the intensity of the hydrological cycle (Sherwood, 2010), climate aridity (Sherwood and Fu, 2014) as well as severity of drought events (Rebetez et al., 2006; Marengo et al., 2008).

In a changing climate, temperature rise, as suggested by different climate scenarios, may impact the atmospheric humidity. According to the Classius-Clapeyron (CC) relationship, a temperature rise of 1 ℃ is sufficient to increase the equilibrium amount of water vapor of the air by roughly 7%. Given the unlimited water availability in the oceans as well as the projected temperature rise, water vapor content is expected to increase, at least in the oceanic areas, in order to maintain RH constant in the future. Particularly, there is empirical evidence on the increase in the water vapor content at both the surface and upper tropospheric levels (Trenberth et al., 2005). In this context, numerous studies have supported the constant RH scenario under global warming conditions (e.g. Dai, 2006; Lorenz and Deweaver 2007; Willett et al., 2008; McCarthy et al., 2009; Ferraro et al., 2015). In contrast, other studies supported the non-stationary behavior of RH, not only in continental areas located far from oceanic humidity (e.g. Pierce et al., 2013), but also in humid regions (e.g. Van Wijngaarden and Vincent, 2004). Assuming the stationary behavior of RH, the influence of RH on AED may be constrained, given that any possible change in AED would be mostly determined by changes in other aerodynamic variables (e.g. air temperature and wind speed) (McVicar et al., 2012a and b) or by changes in cloudiness and solar radiation (Roderick and

Farquhar, 2002; Fan and Thomas, 2013). However, a range of studies have supported

the non-stationary behavior of RH under global warming, giving insights on significant

changes in RH over the past decades. A representative example is Simmons et al.

(2010) who compared gridded observational and reanalysis RH data, suggesting a clear

dominant negative trend in RH over the Northern Hemisphere since 2000. Also, based

on a newly developed homogeneous gridded database that employed the most available

stations from the telecommunication system of the WMO, Willett et al. (2014) found

significant negative changes in RH, with strong spatial variability, at the global scale.

This global pattern was also confirmed at the regional scale, but with different signs of

change, including both negative (e.g. Vincent et al., 2007; Vicente-Serrano e al., 2014b;

2016; Zongxing et al., 2014) and positive trends (e.g. Shenbin, 2006; Jhajharia et al.,

2009; Hosseinzadeh Talaee et al., 2012).

There are different hypotheses that explain the non-stationary evolution of RH under

global warming conditions. One of these hypotheses is related to the slower warming of

oceans in comparison to continental areas (Lambert and Chiang, 2007; Joshi et al.,

2008). In particular, specific humidity of air advected from oceans to continents

increases more slowly than saturation specific humidity increases over land (Rowell and

Jones 2006; Fasullo 2010). This would decrease RH over continental areas, inducing an

increase in AED and aridity conditions (Sherwood and Fu, 2014). Some studies

employed global climate models (GCMs) to support this hypothesis under future

warming conditions (e.g. Joshi et al., 2008; O'Gorman and Muller, 2010; Byrne and

O'Gorman, 2013). However, empirical studies that support this hypothesis using

observational data are unavailable. Moreover, the observed decrease in RH over some

coastal areas, which are adjacent to their sources of moisture, adds further uncertainty to

this hypothesis (Vicente-Serrano et al., 2014b and 2016; Willet et al., 2014).

Another hypothesis to explain the non-stationary evolution of RH is associated with land-atmosphere feedback processes. Different studies indicated that atmospheric moisture and precipitation are strongly linked to moisture recycling in different regions of the world (e.g. Rodell et al., 2015). Thus, evapotranspiration may contribute largely to water vapor content and precipitation over land (Stohl and James, 2005; Bosilovich and Chern, 2006; Trenberth et al., 2007; Dirmeyer et al., 2009; van der Ent et al., 2010). Land-atmospheric feedbacks may also have marked influence on atmospheric humidity (Seneviratne et al., 2006); given that soil drying can suppress evapotranspiration, reduce RH and thus reinforce AED. All these processes would again reinforce soil drying (Seneviratne et al., 2002; Berg et al., 2016).

Indeed, it is very difficult to determine which hypothesis can provide an understanding of the observed RH trends at the global scale. Probably, the two hypotheses combined together can be responsible for the observed RH trends in some regions of the world (Rowell and Jones, 2006). In addition to the aforementioned hypotheses, some dynamic forces, which are associated with atmospheric circulation processes, can explain the non-stationary behavior or RH worldwide (e.g. Goessling and Reick, 2011). However, defining the relative importance of these physical processes in different world regions is quite challenging (Zhang et al., 2013; Laua and Kim, 2015).

The objective of this study is to compare the recent variability and trends of RH with changes in the two types of fluxes that affect RH: i) vertical fluxes that were assessed using land evapotranspiration and precipitation and ii) advection that was quantified using oceanic evaporation from moisture source areas. The novelty of this work stems from the notion that although different studies have already employed GCM's and different scenarios to explain the possible mechanisms behind RH changes under warming conditions, we introduce a new empirical approach that employs different

observational data sets, reanalysis fields and a lagrangian-based approach, not only for
identifying the continental and oceanic moisture areas for different target regions, but
also for exploring the relevance of the existing hypothesis to assess the magnitude, sign
and spatial patterns of RH trends in the past decades at the global scale.

**2. Data and methods**
2.1. Dataset description
*2.1.1. Observational RH dataset*
We employed the monthly RH HadISDH dataset, available through
http://www.metoffice.gov.uk/hadobs/hadisdh/. This dataset represents the most
complete and accurate global dataset for RH, including observational data from a wide
range of stations worldwide (Willet et al., 2014). Given that HadISDH includes some
series with data gaps; our decision was to choose only those series with no more than
20% of missing values over the period 1979-2014. In order to fill these gaps, we created
a standardized regional series for each station using the most correlated series with each
target series. While this procedure maintains the temporal variance of the original data,
it provides a low biased estimation of the missing values. In order to avoid biases,
mostly originated from differences in the distribution parameters (mean and variance)
between the candidate and the objective data series, a bias correction was applied to the
candidate data. Thus, normal distribution was used for bias correction of RH. The data
of the candidate series were re-scaled to match the statistical distribution of the
observed series to be filled, based on the overlapping period between them. Overall, a
final dataset of 3462 complete stations spanning different regions worldwide and
covering the period 1979-2014 was employed in this work.

*2.1.2. Reanalysis RH dataset*
Daily data of dewpoint ($T_d$), air temperature (T) and surface pressure ($P_{mst}$) at a spatial
interval of 0.5º was obtained from the ERA-Interim, covering the period 1979-2014
(http://www.ecmwf.int/en/research/climate-reanalysis/era-interim) (Dee et al., 2011). To
calculate RH, we followed the formulation used by Willett et al. (2014) for the
HadISDH RH dataset. The reason for this is to make better comparisons between RH
obtained from observations in the HadISDH and those derived from the ERA-Interim
dataset. Based on the selected variables, we calculated the daily RH following Buck

149     (1981):

$$RH = 100 \left( \frac{e}{e_s} \right) \qquad (1)$$

where $e$ is the actual vapor pressure in hPa and $e_s$ is the saturated vapor pressure in hPa.
As a function of the wet bulb air temperature ($T_w$) in ºC, $e$ is estimated following two
different equations with respect to water/ice. If $T_w$ is above 0ºC, $e$ is calculated as :
$$e = 6.1121 \cdot f_w exp \left( \frac{\left( 18.729 - \left[ \frac{T_d}{227.3} \right] \right) \cdot T_d}{257.78 + T_d} \right) \qquad (2)$$

Where $T_d$ is the dew point temperature in ºC
If $T_w$ is below 0ºC, $e$ it is calculated as:
$$e = 6.1115 \cdot f_i exp \left( \frac{\left( 23.036 - \left[ \frac{T_d}{333.7} \right] \right) \cdot T_d}{279.82 + T_d} \right) \qquad (3)$$

where
$f_w = 1 + 7 \times 10^{-4} + 3.46 \times 10^{-6} P_{mst}$     (4)
$f_i = 1 + 3 \times 10^{-4} + 4.18 \times 10^{-6} P_{mst}$     (5)
Where $P_{mst}$ is the pressure at the height level.
$T_w$ is obtained according to Jensen et al. (1990):
$T_w = \frac{aT + bT_d}{a + b}$     (6), where
$a = 6.6 \times 10^{-5} P_{mst}$     (7)
$b = \frac{409.8e}{(T_d + 237.3)^2}$     (8)
and T is the 2 meters air temperature in ºC
$e_s$ is obtained by substituting $T_d$ by T .
*2.1.3. Land precipitation and land air temperature*
We employed the gridded land precipitation and surface air temperature data (TS
v.3.23), provided by the Climate Research Unit (UK), at a 0.5º spatial interval for the
period 1979-2014 (Harris et al., 2014). This product was developed using a relatively
high number of observational sites, which guarantees a robust representation of climatic
conditions across worldwide regions. Importantly, this product has been carefully tested
for potential data inhomogenities as well as anomalous data.

*2.1.4. Sea Surface Temperature (SST)*
We used the monthly SST data (HadSST3), compiled by the Hadley Centre for the
common period 1979-2014 (http://www.metoffice.gov.uk/hadobs/hadsst3/). This dataset
is provided at a 0.5º grid interval (Kennedy et al., 2011a and b).

*2.1.5. Ocean evaporation and continental evapotranspiration data*
To quantify the temporal variability and trends of land evapotranspiration and oceanic
evaporation, we employed two different datasets. First, the oceanic evaporation was
quantified using the Objectively Analyzed air-sea Fluxes (OAFLUX) product (Yu et al.,
2008) from 1979 to 2014, which was used to analyze recent variability and changes in
evaporation from global oceans (Yu, 2007). To account for land evapotranspiration, we
employed the Global Land Evaporation Amsterdam Model (GLEAM) (Version 3.0a)
(http://www.gleam.eu/) (Miralles et al., 2011) from 1980 to 2014. This data set has been
widely validated using in situ measurements of surface soil moisture and evaporation
across the globe (Martens et al., 2016).

  2.2. Methods

*2.2.1. Relative Humidity (RH) trends*
We assessed the seasonal (boreal cold season: October-March; boreal warm season:
April-September) and annual trends of RH for 1979-2014 using two different global
datasets (HadISDH and ERA-Interim). To quantify the magnitude of change in RH, we
used a linear regression analysis between the series of time (independent variable) and
RH series (dependent variable). The slope of the regression indicates the amount of
change (per year), with higher slope values indicating greater changes. To assess the
statistical significance of the detectable changes, we applied the nonparametric Mann–
Kendall statistic, which measures the degree to which a trend is consistently increasing
or decreasing (Zhang et al., 2001). To account for any possible influence of serial
autocorrelation on the robustness of the defined trends, we applied the modified Mann–
Kendall trend test, which returns the corrected p-values after accounting for temporal
pseudoreplication in RH series (Hamed and Rao, 1998; Yue and Wang, 2004). The
statistical significance of the trend was tested at the 95% confidence level ($p<0.05$).
Following the trend analysis results, we selected those regions that showed a high
agreement between HadISDH and ERA-Interim datasets in terms of the sign and
magnitude of RH changes. Nevertheless, we also extended our selection to some other
regions, with low station density in the HadISDH dataset. This decision was simply
motivated by the consistent changes found over these regions, as suggested by the ERA-
Interim dataset. For all the defined regions, we identified the oceanic and continental
moisture sources by means of the FLEXPART lagrangian model.

*2.2.2. Identification of continental and oceanic moisture sources*
We used the FLEXPART V9.0 particle dispersion model fed with the ERA-Interim
reanalysis data. According to this model, the atmosphere is divided homogeneously into

three-dimensional finite elements (hereafter "particles"); each represents a fraction of

the total atmospheric mass (Stohl and James, 2004). These particles may be advected

backward or forward in time using three-dimensional wind taken from the ERA-Interim

data every time step, with superimposed stochastic turbulent and convective motions.

The rates of increase (e) and decrease (p) of moisture (e-p) along the trajectory of each

particle were calculated via changes in the specific moisture (q) with time (e-p =

mdq/dt), where m is the mass of the particle. Similar to the wind field, q is also taken

from the meteorological data. FLEXPART allows identifying the particles affecting a

particular region using information about the trajectories of these selected particles. A

description of this methodology is detailed in Stohl and James (2004).

The FLEXPART dataset used in this study was provided by a global experiment in

which the entire global atmosphere was divided into approximately 2.0 million

"particles". The tracks were computed using the ERA-Interim reanalysis data at 6 h

intervals, at a 1° horizontal resolution and at a vertical resolution of 60 levels from 0.1

to 1000 hPa. For each particular target region, all the particles were tracked backward in

time, and its position and specific humidity (q) were recorded every 6 h. With this

methodology, the evaporative sources and sink regions for the particles reaching the

target region can be identified. All areas where the particles gained humidity (E -P > 0)

along their trajectories towards the target region can be considered as "sources of

moisture". In contrast, all areas with lost humidity (E -P < 0) are considered as "sinks".

A typical period used to track the particles backward in time is 10 days that is the

average residence time of water vapor in the global atmosphere (Numaguti, 1999).

However, we followed the methodology of Miralles et al (2016), where an optimal

lifetime of vapor in the atmosphere was calculated to reproduce the sources of moisture.

As such, three steps were carried out in this order: i) all the particles that leave each

target region were tracked back during 15 days and the "initial sources" at annual scale

were defined as those areas with positive (E-P) values, ii) from these "initial sources",

all the particles were forward tracked during 1 to 15 days individually, and (E-P)<0 was

calculated for these lifetime periods to estimate the precipitation contribution over the

target region, iii) the optimal lifetime selected for each region was chosen according to

the minimum absolute difference between the FLEXPART simulated precipitation and

the CRU TS v.3.23 for each region, iv) and finally the backward tracking was

recalculated during these optimal lifetimes.

We defined the climatological spatial extent of each source region corresponding to a

particular target region by applying a 95[th] percentile criterion computed for the annual

and seasonal (boreal summer and winter) positive (E-P) field (Vazquez et al., 2016).

Then, for each year of the period, we estimated the total moisture supply from each

source region.

Also from FLEXPART simulations, we obtained the fractions of moisture from the

continental and oceanic sources annually and for each cold and warm season. The

purpose was to compare with the results obtained on the role of the land

evapotranspiration and ocean evaporation of RH variability and trends.

*2.2.3. Relationship between RH and the selected land/oceanic climate variables*

Based on defining the spatial extent of each moisture source region, we calculated

annual, warm and cold season regional series for ocean evaporation and land

evapotranspiration using the OAFLUX and GLEAM datasets, respectively. The

regional series of ocean evaporation and land evapotranspiration were created using a

weighted average based on the seasonal/annual fields of (E-P)>0 (Section 2.2.2). This

approach allows creating a time series that better represents the interannual variability

of ocean evaporation and land evapotranspiration in the source(s) of moisture for each
defined region. Following the same approach, we also calculated the regional series of
SST corresponding to each oceanic moisture source region. Likewise, we calculated the
regional series of land precipitation and air temperature for each target region using
CRU TS v.3.23 dataset, and the ratio between air temperature in the target region and
SST in the source region.
For each target region, we related the regional series of seasonal and annual RH to the
corresponding regional times series of all aforementioned climatic variables. However,
to limit the possible influence of the trends presented in the data itself on the computed
correlations, we de-trended the series of the climate variables prior to calculating the
correlation. We also assessed changes in the regional series of the different variables;
their statistical significance was tested by means of the modified Mann-Kendall test at
the 95% level. For each target region, we summarized the results of the magnitude of
change in RH as well as other investigated variables at the seasonal and annual scales.
However, to facilitate the comparison among the different variables and the target
regions worldwide, we transformed the amount of change of each variable to
percentages.
Finally, we also computed the association between RH and land evapotranspiration at
the annual and seasonal scales using the available gridded evapotranspiration series.
While a pixel-to-pixel comparison does not produce a reliable assessment of the
possible contribution of land evapotranspiration to RH changes, given that the source of
moisture can apparently be far from the target region, we still believe that this
association can give insights on the possible relationship between land
evapotranspiration on RH changes.

**3. Results**

**3.1. Trends in Relative Humidity**

Figure 1 shows the average seasonal and annual RH and the Vertically Integrated Moisture Flux (VIMF), which can be used to estimate regions where the precipitation dominates (negative values) over the evaporation (positive values), from the ERA-Interim dataset. RH shows higher average values over equatorial regions, Southeast Asia and the North Eurasia region. The lower values are recorded over tropical regions, mainly in the North Hemisphere. Spatial differences between the cold and warm seasons are very low. The annual pattern of the VIFM over continents shows that precipitation exceeds evaporation over the Intertropical Convergence Zone, Southeast Asia and the islands between the Pacific and Indian Oceans (Maritime continent), a great part of South America, Central America, Central Africa, and northward to 40ºN in the Northern Hemisphere. Evaporation is higher than precipitation over the main area of Australia, the Pacific coast of North America, Northeast Brazil, areas around the Mediterranean Sea, eastern coast of Africa and southwest Asia. Seasonally it is evident the poleward movement of the ITCZ during the hemispheric summer, and the change of the pattern over North America and Eurasian continent.

Figure 2 summarizes the magnitude of change in RH for the boreal cold and warm seasons and at the annual scale, calculated using the annual and seasonal (boreal summer and winter) for the period between 1979 and 2014. For HadISDH, it is noted that the available RH stations is unevenly distributed over the globe, with higher density in the mid-latitudes of the Northern Hemisphere. Nevertheless, the available stations show coherent and homogeneous spatial patterns of RH changes (Supplementary Figure 1). In the boreal cold season, the most marked decrease was observed in the Southwest and areas of Northeast North America, central Argentina, the Fertile Crescent region in

western Asia, Kazakhstan, as well as in the eastern China and the Korea Peninsula. On the other hand, the dominant RH increase was recorded in larger areas, including most of Canada (mostly in the Labrador Peninsula), and large areas of North and central Europe and India. While the density of complete and homogeneous RH series is low, we found a dominant positive trend across the western Sahel and South Africa. The ERA-Interim dataset showed magnitudes of change close to those suggested by HadISDH. In addition, the ERA-Interim also provides information on RH changes in regions with low density of RH observations (e.g. East Amazonia, east Sahel and Iran), suggesting a dominant RH decrease across these regions.

For the boreal warm season, a clear tendency towards a reduction in RH was observed in vast regions of the world, including (mostly the Iberian Peninsula, France, Italy, Turkey and Morroco), Eastern Europe, and western part of Russia. Based on the available stations across central Asia, we also found a general reduction of RH; a similar pattern was also observed in East Asia, including Mongolia, east China, north Indonesia, South Japan and Korea. This reduction was also noted South America, with a general homogeneous pattern over Peru, Bolivia and a strong decrease over central Argentina. On the other hand, the positive evolution of RH observed during the cold season across Canada and Scandinavia was reinforced during the boreal warm season. In the Western Sahel and India, we found an upward trend of RH. The ERA-Interim also revealed a strong RH decrease over the whole Amazonian region and the West Sahel, while a marked increase dominated over the Andean region between Colombia, Ecuador and North Peru. In Australia, the spatial patterns were more complex than those obtained using the available observatories.

The HadISDH dataset suggests a general decrease of RH over Southwest North America, Argentina, central Asia, Turkey, Mongolia and China, with a particular

reduction over the Eastern Sahel, Iran, Mongolia and the eastern Asia. On the other
hand, a dominant positive trend was observed across Canada, areas of North Southern
America, the Western Sahel, South Africa (Namibia and Botswana), some areas of
Kenia, India and the majority of Australia. A wide range of these regions exhibited
statistically significant trends from 1979 to 2014. (Supplementary Figure 2). A
statistically significant negative trend was observed at the seasonal and annual scales,
not only in most of Southern America and Northern America, but in large regions of
Africa, South Europe, central and East Asia as well. On the other hand, areas of
complex topography in the Northern Hemisphere, Australia, India, Northern South
America and Africa showed positive trends.
Albeit with these complex spatial patterns of RH changes, there is a global dominant
negative trend (Figure 3). This pattern was observed using both the HadISDH and the
ERA-Interim datasets, although there is marked spatial bias in data availability of the
HadISDH. Figure 4 illustrates the relationship between the magnitudes of change in
RH, as suggested by the HadISDH dataset versus the ERA-Interim dataset. At the
seasonal and annual scales, there is a relatively high correlation (mostly above 0.55).
Given this high consistency between the HadISDH and the ERA-Interim datasets in
terms of both the magnitude and sign of change in RH (Figures 2 and 3) and also in the
interannual variations (Supplementary Figures 3 and 4), we decided to restrict our
subsequent analysis to the ERA-Interim dataset, recalling its denser global coverage
compared to the HadISDH.
As RH is mostly dependent on changes in specific humidity (q), there is a dominant
high correlation between the interannual variability of RH and q (Supplementary Figure
5). In accordance, the magnitude of observed change in these two variables showed a
strong agreement for 1979-2014. Figure 5 summarizes the magnitude of change in
specific humidity (q) as well as changes in specific humidity necessary to maintain RH
constant as recorded in 1979. Specific humidity showed the strongest decrease in
Southwest North America, the Amazonian region, Southern South America and the
Sahel regions: a spatial pattern that is similar to RH pattern. Given the evolution of air
temperature between for 1979-2014, these regions exhibited a deficit of water vapor on
the order of -2 g/kg$^{-1}$ in order to maintain RH constant.

**3.2. Spatial patterns of the dependency between RH and climate variables**
Based on the high agreement between the HadISDH and the ERA-Interim datasets in
reproducing consistent seasonal and annual trends in RH, we selected a range of regions
(N=14) worldwide (Figure 6). For these selected regions, we assessed the connection
between RH and some relevant climatic variables for the period 1979-2014. In addition,
we defined the oceanic and continental sources of moisture corresponding to these
regions using the FLEXPART model. We assessed the optimal lifetime for each region:
during 4 days in back for regions 1-5 and 7-11, during 5 days for regions 6, 12-13, and
during 7 days for region 5 (see section 2.2). Figures 7-9 show some examples of the
dependency between RH and different climate variables at the annual scale. Results for
all regions at the seasonal and annual scales are presented in supplementary materials.
***3.2.1. Western Sahel***
Figure 7 (top) illustrates RH trends in the Western Sahel using the HadISDH and ERA-
Interim datasets. We also showed the distribution of the average annual moisture
sources (E-P in mm) over this region for 1979-2014. As illustrated, the atmospheric
moisture is mostly coming from the western Sahel region itself, in addition to some
oceanic sources located in the central eastern Atlantic Ocean. At the seasonal scale,
there are some differences in the location and the intensity of the moisture sources, with
more oceanic contribution during the boreal warm season. However, in both cases, the
continental moisture seems to be the key source of humidity in the region (Suppl.
Figures 21 and 35). In other areas, e.g. the Western European region (Suppl. Figures 17
and 31), we observed marked differences in the location and the intensity of humidity
sources between the boreal cold and warm seasons. Figure 7 (central) shows different
scatter plots summarizing the relationships between the de-trended annual series of RH
and those of relevant climate variables (e.g. precipitation, air temperature and SST). As
illustrated, the interannual variability of RH in the region is correlated to changes in the
total annual precipitation and the total annual land evapotranspiration in the continental
source region. Specifically, the correlation between the de-trended annual RH and
precipitation and land evapotranspiration is generally above 0.8 (p < 0.05). In contrast,
RH shows negative correlations with air temperature and SST ratio over the oceanic
source. While the correlation is statistically insignificant (p>0.05), it suggests that
higher differences between air temperature and SST reinforce lower annual RH. At the
seasonal scale, we found similar patterns (Supplementary Figs. 21 and 35), with RH
being highly correlated with land evapotranspiration during the boreal cold and warm
seasons. Nevertheless, in the warm season, a significant negative correlation with air
temperature and SST ratio was observed. This pattern concurs with the significant
increase in specific humidity (q) for 1979-2014; this is probably related to the high
increase in land evapotranspiration (19.5%, p < 0.05).

### 414   *3.2.2. La Plata region*

Figure 8 summarizes the corresponding results, but for La Plata region (South
America). Results indicate a general decrease in RH at the annual and seasonal scales
using both the HadISDH observational data and the ERA-Interim dataset. As depicted,
the main humidity sources are located in the same region, combined with some other
continental neighbor areas over South America. A similar finding was also observed on
the seasonal scale (Supplementary Figs. 25 and 39). Similar to the West Sahel region,
we found a significant association between the interannual variations of RH and
precipitation and the land evapotranspiration in the continental source region. Similarly,
we did not find any significant correlation between RH changes and the interannual
variability of the oceanic evaporation in the oceanic source region as well as the ratio
between air temperature in the continental target region and SST in the oceanic source
region. Again, we found a negative correlation between RH and air temperature/SST
ratio, though being statistically insignificant at the annual scale ($p>0.05$). In La Plata
region, we noted a strong decrease in RH (-6.21%/decade) for 1979-2014, which agrees
well with the strong decrease in absolute humidity. This region is strongly impacted by
continental atmospheric moisture sources, with a general decrease in precipitation and
land evapotranspiration during the analyzed period.

### *3.2.3. Southwest North America*
Results for Southwest North America are also illustrated in Figure 9. In accordance with
both previous studied examples (West Sahel and La Plata), this region also exhibited a
strong and positive relationship between the interannual variability of RH and
precipitation and land evapotranspiration. This pattern was also recorded for the boreal
warm and cold seasons (Supplementary Figures 28 and 42). In this region, we found a
strong negative trend of RH for 1979-2014, which concurs with the significant decrease
of absolute humidity. We noted a significant increase in air temperature, air temperature
and air temperature to SST ratio, while a negative and statistically significant decrease
in land evapotranspiration in the continental sources of moisture was observed.

### *3.2.4. Other regions*

Other regions of the world (see Supplementary Material) also showed strong dependency between the interannual variability of RH and that of land evapotranspiration in the land moisture sources. Some examples include Western Europe, Central-eastern Europe, Southeast Europe, Turkey, India and the east Sahel. Nevertheless, the influence of land evapotranspiration was very different between the boreal warm and cold seasons (e.g. Scandinavia, Central-East Europe and the Amazonian region). In contrast, other regions showed a weak correlation between the temporal variability of RH and land evapotranspiration in the moisture source region. A representative example is China, which witnessed a strong decrease in RH for 1979-2014. This might be explained largely by the fact that relative interannual ET variations are just much weaker in China compared to other regions so that the signal-to-noise ratio is worse in China. In this region, RH changes correlated significantly with annual precipitation only: a variable that did not show significant changes from 1979 to 2014 (Supplementary Fig. 11). This annual pattern was also observed for the boreal cold and warm seasons (Supplementary Figs. 23 and 37).

Nevertheless, although the interannual variability of land evapotranspiration in the land moisture sources showed the highest correlation with RH variability in the majority of the analyzed regions, air temperature/SST ratio in the oceanic moisture sources also exhibited negative correlations with RH in particular regions, including West Sahel, La Plata, West Coast of the USA, Central-Eastern Europe, India, central North America and the Amazonian region. This finding suggests that higher differences between air temperature in the target area and SST in the oceanic moisture region would favor decreased RH.

468

**3.3. Land and ocean contribution to RH trends**

It is well-recognized that establishing a direct influence of land evapotranspiration on RH is a challenging task, including also any attempt to directly compare these influences with the possible contribution from oceanic evaporation and moisture transport. This is primarily because, apart from very humid regions, the increase in land evapotranspiration could be driven by increased precipitation, which is accompanied by anomalous RH conditions. This dependency explains the correlation found between precipitation and land evapotranspiration in some regions worldwide, although there are considerable spatial and seasonal differences (Supplementary Figures 45 to 47). In cold and humid regions, land evapotranspiration is also related to the interannual variability of the AED (Supplementary Figures 48 to 50). Correspondingly, the magnitude of the oceanic evaporation may be insufficient to explain RH anomalies in the target region. Taken together, the transport of moisture to any target region is a fundamental process, in which atmospheric circulation configurations can contribute significantly to RH anomalies and their spatial variations. Hence, for better understanding of the possible contribution of oceanic evaporation and land evapotranspiration to RH, we assessed the contribution of land and ocean to precipitation, as represented by (E-P). It may provide some clues on the possible connection between oceanic moisture and RH variability. Figures 9 and 10 illustrate the relationship between the interannual variability of RH in each target region and land-oceanic contribution to the annual precipitation. . Overall, results reveal important differences among the analyzed regions, with  statistically insignificant correlations found between the interannual variations of RH in some regions ocean and land contribution to precipitation. One example is the positive and significant correlation found between the annual RH and the ocean E-P in regions 2

(Scandinavia), 5 (Western Sahel), 6 (India), 12 (West North America) and 13
(Amazonia). A similar pattern was observed at the seasonal scale, albeit with greater
contribution during the cold season, especially in the regions where precipitation is
mostly driven by western flows during this season (e.g. West North America, Western
Europe and Scandinavia) (Supplementary Figs 51 to 54). On the other hand, the
contribution of land areas to precipitation in the analyzed target regions is rather
complex, with strong spatial differences. At the annual scale, a positive and significant
contribution to precipitation (E-P) is found in regions 3 (Central-East Europe), 6 (India),
7 (China), 9 (La Plata) and 11 (central US). Notably, a wide range of regions exhibit a
positive and significant correlation between land (E-P) and RH during the cold season,
including Scandinavia, Central-East Europe, South-East Europe and Turkey, Western
Sahel, India, North-East Asia, La Plata, Central USA, West North America and East
Sahel. In contrast, during summertime, only regions 3 (Central-East Europe) and 9 (La
Plata) show a significant correlation between land contribution to precipitation (E-P)
and annual RH. Given these noticeable spatial and seasonal differences, our findings
suggest that there are no generalized patterns in terms of the contribution of ocean and
land to the interannual variability of RH. This complexity makes it quite difficult to
attribute RH trends between 1979 and 2014 at the global scale to a unique driver or
process.. The FLEXPART model outputs do not suggest consistent trends for ocean as
well as land E-P. Irrespective of all these limitations, we believe that it is still possible
to assess with a degree of confidence the evolution of land and ocean contribution (E-P)
to precipitation of the target regions. Figure 12 illustrates the evolution of the land
contribution (E-P) to precipitation in the different target regions. We noted positive and
significant changes in the region 2 (Scandinavia), 3 (Central-East Europe), 5 (Western
Sahel) and 14 (Eastern Sahel). A contradictory behavior is observed for region 9 (La
Plata), with a statistically significant downward trend in land contribution (E-P) to
precipitation. At the seasonal scale, results suggest considerable differences
(Supplementary Figures 54 and 55), with no clear positive or negative trends.
With respect to the different analysed variables, changes in RH were more associated
with those of land evapotranspiration across the selected regions (Figure 13). In
contrast, changes in annual RH did not correlate significantly with the observed changes
in precipitation, air temperature/SST and oceanic evaporation. The observed patterns
were similar for both the warm and the cold season (Supplementary Figs. 56 and 57).
Indeed, these positive and significant correlations do not imply causation between these
factors and RH variations over space and time. . This is simply evident in different
regions worldwide, where there are strong seasonal and spatial differences in the
contribution of land evapotranspiration to RH. Nevertheless, these findings also confirm
the role of land evapotranspiration in explaining the observed variability of RH.
Specifically, for many regions and at different temporal scales (i.e. seasonal and
annual), changes in land contribution to precipitation show statistically significant
positive correlation with changes in evapotranspiration and precipitation
(Supplementary Figure 58). Again, this correlation does not imply a true causal
relationship between RH variability and evapotranspiration, given the strong coupling
between many of these controlling variables (e.g. precipitation, RH, and land
evapotranspiration). This coupling is also spatially and temporarily variable. However,
this good agreement between changes in the land contribution to precipitation and
changes in land evapotranspiration emphasizes the role of land evapotranspiration in
explaining the complex spatial patterns of RH changes, In many regions, land
contribution (E-P) to precipitation is evidently important, with contributions close to
50% or even higher, including the  Western Sahel (54%), Eastern Sahel (61%) and
North East Asia (64%) (Supplementary Table 1).
Nevertheless, there is some uncertainty in attributing RH changes to land
evapotranspiration. Figure 14 depicts the relationship between RH and land
evapotranspiration seasonally and annually at the global scale. Note that these are local
("pixel-by-pixel") correlations and the interpretation differs from the previous analysis
where RH in target regions is correlated with ET in corresponding source regions.
Results reveal strong positive and significant correlations in large areas of the world.
The strongest positive correlations were found in Central, West and Southwest North
America, Argentina, east Brazil, South Africa, the Sahel, central Asia and the majority
of Australia. Nevertheless, there are some exceptions, including large areas of the
Amazon, China, central Africa and the high latitudes of the Northern Hemisphere,
where the correlations were negative. In general, the areas with positive and significant
correlations between RH and land evapotranspiration corresponded to those areas
characterized by semiarid and arid climate characteristics, combined with some humid
areas (e.g. India and northwest North America). Nevertheless, at the global scale, the
correlation between RH and land evapotranspiration shows spatial patterns consistent
with those based on the correlation between RH and precipitation. , Similar spatial
patterns are found also for the correlation between precipitation and  evapotranspiration
at both seasonal and annual scales (Supplementary Figures 59 and 60), This high
agreement makes it difficult to accurately define the most dominant variable (s), which
control the temporal variability of RH, given the good spatial agreement between RH
and different variables (e.g. precipitation, land evapotranspiration).  The complex spatial
patterns of the observed trends for different variables add another source of uncertainty
to proper attribution of RH changes. Figure 15 illustrates the spatial distribution of the

magnitude of change in annual and seasonal land evapotranspiration at the global scale

from 1979 to 2014. As depicted, the spatial patterns of land evapotranspiration changes

resemble those of RH in some regions (refer to Figure 2). For example, a positive trend

in the annual land evapotranspiration dominated over the Canadian region, which agrees

well with the general increase in RH across the region. On the other hand, there was a

dominant decrease in the annual land evapotranspiration across vast areas of North

America, which concurs also with the strong decrease in RH. Similar to the pattern

observed for land evapotranspiration, RH increased particularly over southwest North

America. In South America, both variables also showed a dominant negative trend at

the annual scale, but with some spatial divergences, mainly in the Amazonian region.

Specifically, the western part of the basin showed the most important decrease in land

evapotranspiration, whereas the most significant decrease in RH was observed in the

eastern part. In the African continent, some areas showed good agreement between RH

and land evapotranspiration changes, in terms of both the sign and magnitude. This can

be clearly seen in the West and East Sahel, where a strong gradient in RH trend between

the West (positive) and the East (negative) was observed. Nevertheless, other African

regions showed a divergent pattern between both variables. One example is the Guinea

Gulf in Nigeria and Cameroon, where we noted a strong increase in land

evapotranspiration, as opposed to RH changes. In Australia, although both variables

showed a dominant positive trend, they did not match exactly in terms of the spatial

pattern of the magnitude of change. The Eurasian continent showed the main

divergences between both variables. In the high latitudes of the continent, there was a

dominant increase in both variables. For other regions (e.g. Western Europe), we noted

a dominant RH decrease, which was not observed for land evapotranspiration. A similar

pattern was observed over east China, with a dominant RH negative trend and a positive

land evapotranspiration. Overall, the lack of significant spatial association between the
magnitude of trends in RH and the magnitude of trends in evapotranspiration can be
seen in the context of the strong spatial diversity of trends of these two variables at both
annual and seasonal scales (Supplementary Figure 61). Similar patterns are found also
for the trends of RH and precipitation, and the trends of precipitation and land
evapotranspiration, although precipitation trends show a very different pattern, in
comparison to land evapotranspiration trends (Supplementary Figure 62).  Results
suggest that while the variability of precipitation, RH and land evapotranspiration show
strong interannual associations, their observed trends are completely decoupled over
space. This high spatial variability of trends at the global scale confirms that direct
attribution of observed RH changes to oceanic/land contributions is a challenge and
quite complex task.
In relation to the influence of the ocean evaporation, our results confirm that the global
connection between oceanic evaporation and changes in RH is also complex. On one
hand, it is difficult to establish a pixel per pixel relationship since the sources of
moisture may strongly differ at the global scale. On the other hand, it is not feasible to
identify moisture sources for each 0.5º pixel at the global scale. However, we believe
that the analysis of the evolution of SST and oceanic evaporation for 1979-2014 and the
evolution of the oceanic contribution to precipitation can give indications on some
relevant patterns. Figure 16 illustrates the spatial distribution of the magnitude of
change of annual and seasonal SST and oceanic evaporation. Supplementary Fig. 63
shows the spatial distribution of trend significance. As depicted, complex spatial
patterns and high variability of the trends were observed, particularly for oceanic
evaporation. Furthermore, the spatial distribution of the magnitude of change in annual
and seasonal oceanic evaporation was not related to the SST changes (Supplementary

Fig. 64). This finding suggests that oceanic evaporation is not only driven by changes in SST. Thus, although some regions showed positive changes in the oceanic evaporation, the amount of increase was much lower than that found for SST, which suggests that only SST changes do not drive evaporation changes (Supplementary Figure 65, Supplementary Table 2).

## 4. Discussion

### 4.1. Relative Humidity trends

We assessed the temporal variability and trends of relative humidity (RH) at the global scale using a dense observational network of meteorological stations (HadISDH) and reanalysis data (ERA-Interim). Results revealed high agreement of the interannual variability of RH using both datasets for 1979-2014. This finding was also confirmed, even for the regions where the density of the HadISDH observatories was quite poor (e.g. the northern latitudes and tropical and equatorial regions). Recent studies have suggested dominant decrease in observed RH during the last decade (e.g. Simmons et al., 2010; Willet et al., 2014). Our study suggests dominant negative trends of RH using the HadISDH dataset. This decrease is mostly linked to the temporal evolution of RH during the boreal warm season. Nevertheless, other regions showed positive RH trends. In accordance with the HadISDH dataset, the ERA-Interim revealed dominant negative RH trends, albeit with a lower percentage of the total land surface compared to the HadISDH dataset. These differences cannot be attributed to the selected datasets, given that both mostly agree on the magnitude and sign of changes in RH.

Observed changes in RH were closely related to the magnitude and the spatial patterns of specific humidity changes. Results demonstrate a general deficit of specific humidity to maintain RH constant in large areas of the world, including the central and south

Northern America, the Amazonas and La Plata basins in South America and the East Sahel. In other regions, RH increased in accordance with higher specific humidity. Some studies suggested that changes in air temperature could partly cancel the effects of the atmospheric humidity to explain RH changes (e.g. McCarthy and Tuomi, 2004; Wright et al., 2010; Sherwood, 2010). Nevertheless, although air temperature trends showed spatial differences at the global scale over the past four decades (IPCC, 2013), our results confirm that air temperature is not the main driver of the observed changes of RH globally. The ERA-Interim dataset clearly showed a close resemblance between RH and specific humidity trends at the global scale. This suggests that specific humidity is the main driver of the observed changes in the magnitude and spatial pattern of RH during the past decades.

### 4.2. Contribution of continental areas to changes in RH

Overall, there is an agreement between the interannual variability of precipitation and land evapotranspiration in the continental moisture source and the interannual variability of RH in different regions. Nevertheless, considering gridded datasets at the global scale, we found that this good agreement is restricted only to the arid and semiarid regions. In humid regions, soil moisture is not a constrained variable; the variability of land evapotranspiration is mostly driven by changes in the AED (Stephenson, 1990). This makes it difficult to unravel the possible direct contribution of land evapotranspiration to the variability of RH using statistical approaches and empirical information, particularly with the strong coupling among these variables. Land evapotranspiration is closely related to precipitation variability in arid and semiarid regions; increased land evapotranspiration thus tends to be caused primarily by increased precipitation, which is accompanied by corresponding RH anomalies. Also,

RH may affect land evapotranspiration, both in arid and humid regions, given its
important contribution to the aerodynamic component of the AED (Wang et al., 2012;
Vicente-Serrano et al., 2014a).
Nevertheless, although the interannual variability of these three variables can be
strongly coupled in some regions, the long-term trends in these variables may strongly
differ, as a consequence of changes in precipitation, increasing influence of AED on
land evapotranspiration, and also changes in land and atmospheric contribution to RH
and precipitation. The fourteen analyzed regions, in which FLEXPART was applied,
show a relevant continental contribution (E-P) to precipitation in these areas. The
average contribution is generally below 40% of the total precipitation in some regions
(e.g. Western Europe, Scandinavia or West North America), which exceeds 50% in
specific regions (e.g. Sahel and East China).
Therefore, it is reasonable to consider that changes in the contribution of continents to
precipitation may affect land evapotranspiration processes and ultimately affect RH
variability in these continental areas. Thus, our results suggest an influence of land-
atmosphere water feedbacks and recycling processes on RH trends. This is simply
because more available soil humidity under favorable atmospheric and land conditions
would result in more evapotranspiration and accordingly higher air moisture (Eltahir
and Bras, 1996; Domínguez et al., 2006; Kunstmann and Jung, 2007). Recalling that the
ocean surface evaporates about 84% of the water evaporated over the Earth (Oki, 2005),
the oceanic evaporation is highly important for continental precipitation (Gimeno et al.,
2010). However, the continental humidity sources can be also more important than
oceanic sources in some regions (e.g. the Sahel) (Wei et al., 2016a). In this context, our
results concur with previous works. For example, numerous model-based studies have
supported an influence of land evaporation processes on air humidity and precipitation
over land surfaces (e.g. Bosilovich and Chern, 2006; Dirmeyer et al., 2009; Goessling
and Reick, 2011). Moisture recycling is strongly important in some regions of the
world, such as China and central Asia, the western part of Africa and the central South
America (Pfahl et al., 2014; van der Ent et al., 2010).
All these studies assessed the role of continental evapotranspiration on average
precipitation conditions, with few studies focusing on the possible impacts of changes in
soil moisture/evapotranspiration on RH. Rowell and Jones (2006) analyzed different
hypotheses to explain the projected summer drying conditions in Europe, suggesting
that soil moisture decline and land–sea contrast in lower tropospheric summer could be
the key factors responsible for this drying. They concluded that reduced evaporation in
summer will drop RH and hence reduced continental rainfall. These would impact soil
moisture and evapotranspiration processes, inducing a reduction in RH and rainfall,
through a range of atmospheric feedbacks. In the same context, the importance of
moisture recycling processes for atmospheric humidity and precipitation has been
recently identified in semi-arid and desert areas of the world (Miralles et al., 2016).
Although our study was limited to specific regions across the world, results indicate that
humidity in the analyzed regions is substantially originated over continents. This
finding concurs with some regional studies that defined sources of moisture (e.g. Nieto
et al., 2014; Gimeno et al., 2010; Drumond et al., 2014; Ciric et al., 2016). Overall, the
spatial differences in the possible attribution of the observed changes in RH to changes
in land evapotranspiration are important. Nevertheless, in some regions where RH is
strongly correlated with land evapotranspiration, there is a significant correlation
between land contributions to RH, suggesting a robust contribution of land processes to
the interannual variability of RH in these regions. A representative example is La Plata
region, where a strong decrease (-6.6% from 1979 to 2014) in RH is suggested by both

observations and reanalysis datasets. This region did not exhibit a significant trend in precipitation, but conversely there is a significant decrease in absolute humidity and land evapotranspiration. In La Plata region, the oceanic evaporation in the source region has shown a significant increase (6.33%) since 1979. However, this increase seems to be insufficient to favor an increase in RH values. Herein, although the average oceanic contribution (E-P) to precipitation is slightly higher (54%), compared to continental contribution (46%), the interannual variability of RH is positively correlated with the interannual variability of land E-P rather than oceanic E-P. Moreover, this region exhibited a significant decrease in land contribution to precipitation between 1979 and 2014.

On the contrary, in other regions like Western US, the large decrease in RH (-6.2%) corresponded to a large decrease in the absolute humidity (-0.58 g/kg). However, it is difficult to attribute this association to changes in land evapotranspiration, given the low (37%) continental contribution to precipitation in these regions. Moreover, Wei et al. (2016b) indicated that the transport of atmospheric moisture from the Pacific is the main contributor to the interannual variability of precipitation in the region. In this sense, we found a strong relationship between the interannual variability of RH in this region and the oceanic E-P, albeit with insignificant trends in the oceanic and continental contribution (E-P) to precipitation in the region. Here, in the absence of significant changes in oceanic evaporation and contribution to precipitation, it could be reasonable to consider that the decrease in absolute humidity is linked to the atmospheric circulation that control moisture transport in the region (Wei et al., 2016b). The decrease in RH could be also due to the positive and significant trend toward higher differences between air temperature in the land target region and the oceanic temperature in the source region.

Therefore, the relationships between RH, land evapotranspiration variability and
changes in the contribution of continental areas to air moisture and precipitation in the
target regions can be extremely complex. Similarly, the relationships between RH and
land evapotranspiration are rather complex and cannot be easily interpreted.
Nonetheless, albeit with the strong uncertainty existing at the global scale, it is
reasonable to consider that strong changes in soil moisture budget, combined with land
evapotranspiration, could impact somehow the complex RH trends observed at the
global scale. In the same context, there is strong evidence that low levels of soil
moisture and land evapotranspiration are usually accompanied by a reinforcement of
low RH, particularly during drought episodes. Under these circumstances, the
suppression of the latent heat flows from the soil to the atmosphere would enhance soil
and vegetation warming and sensible heat, inducing air temperature rise. Also, the lack
of supply of water vapor to the atmosphere favors the decrease of RH and the
reinforcement of severity of heat waves (Hirschi et al., 2011). Seneviratne et al. (2002)
showed that vegetation control on transpiration might contribute significantly to
enhancement of summer drying, particularly when soil water is limited. Other studies
confirmed this finding for other regions worldwide, employing both observational data
(e.g. Hisrchi et al., 2011) and model outputs (e.g. Seneviratne et al., 2006; Fischer et al.,
2007). Our study suggests good spatial agreement between changes in RH and those of
continental contribution to precipitation as well as land evapotranspiration during
summertime. Although this finding is markedly evident for all the analyzed regions, it
should be seen with caution. This is mainly because physical processes driven soil
moisture are more active during the warm season (Vautard et al., 2007 and 2013;
Miralles et al., 2014), which adds difficulty to establish full causality between RH and
other driving forces during this season.

*4.2. Contribution of oceans to changes in RH*

This study demonstrates strong differences among the fourteen analysed regions in terms of the contribution of Oceanic water bodies to RH variability. In some regions (e.g. Western North America, India, Scandinavia and the Amazonian), the interannual variability of RH is closely related to oceanic E-P, indicating that changes in oceanic evaporation, combined with the processes of atmospheric moisture transport to target regions, play a main role in explaining changes in RH. It is well-recognized that t moisture advection is the main driver of precipitation variability in the majority of world regions (Trenberth, 1999; Wei et al., 2016a). Nevertheless, there is some uncertainty in recent trends of moisture advections from oceanic areas (Zhan and Allan, 2013). As such, it is difficult to determine -in general terms- whether the strong complexity of RH changes identified at the global scale are driven by changes in moisture advections from oceanic areas. Nevertheless, in the context of changing climate, SST, oceanic evaporation, and oceanic contribution to precipitation in the target regions can jointly account for the possible influence of oceans on RH variability in some regions.

Different modelled climate studies suggested strong differences between land and ocean RH trends, as a consequence of the different warming rates between oceanic and continental areas (e.g. Joshi et al., 2008; Dessler and Sherwood, 2009; O'Gorman and Muller, 2010). As the warming rates are generally slower over oceans, the specific humidity of air advected from oceans to continents would increase more slowly than the saturation specific humidity over land, causing a reduction in RH (Rowell and Jones 2006). Due to this effect, RH will not remain constant in areas located very far from humidity sources, as warmer air temperatures under limited moisture humidity would reduce RH (Pierce et al., 2013). This physical process could explain the recent trends of

RH in some regions (e.g. Amazonia), where around 68% of the average atmospheric
moisture originates over oceanic sources. Moreover, RH is correlated with the oceanic
E-P, although there are no changes in the oceanic contribution to precipitation from
1979 to 2014. In Amazonia, RH strongly decreased at the annual scale (-7.7%), as a
consequence of the decrease (-0.56 g/kg) in the absolute humidity, accompanied by an
increase (1.09ºC) in air temperature. While SST in the oceanic source region slightly
increased by 0.33ºC, other variables (i.e. oceanic evaporation, precipitation and land
evapotranspiration) did not t exhibit significant changes. Herein, we believe that the
mechanisms proposed by Sherwood and Fu (2014) to explain decreased RH are
applicable to our study. They attributed RH decrease to sub-saturated oceanic moisture
supply, which compensates a strong air temperature. In this study, this mechanism is
also capable of explaining RH variability, given that the difference in warming rates
between SST in the oceanic moisture source region and air temperature of the
Amazonia is increasing. This feature is also supported by the significant correlation
found between the interannual variability of RH and the ratio between air temperature
and SST.   Again, this explanation does not guarantee full causality between RH
variability and oceanic moisture in the source regions. . A similar pattern is found for
the Eastern Sahel, a region in which continental recycling is particularly important (Wei
et al., 2016a). This region witnessed a strong decrease in RH, but not compensated by
increased precipitation. Although there is no significant correlation between RH and the
ratio between air temperature and SST, the latter variable shows a significant increase.
This could reinforce the drying effect of the suggested trend toward a lower moisture
supply from the oceanic source, especially with the significant negative trend in oceanic
contribution to precipitation in this region. In other regions, there are no coherent
empirical relationships that confirm the impact of oceanic moisture supply processes on
RH changes. Recalling the observed negative RH trend at many coastal regions over the
period 1979-2014, this study confirms that the distance to oceanic humidity sources is
not a key controller of the spatial patterns of RH changes. In many instances, we found
that continental regions, which are very far from oceans (e.g. Canada, central China and
Kazakhstan), recorded a positive RH trend. A possible explanation of these contrasting
findings could be related to the low differences in the warming rates between the
oceanic sources and the majority of the continental target areas. We found that -in most
of the cases- these differences were not strong enough to generate a clear effect at the
global scale, particularly with the available number of observations. In specific regions
(e.g. Amazonia, East Sahel, Western North America), there is a a positive trend in the
difference between air temperature and SST temperature in the source region. These
processes could be the most reasonable physical explanation of the strong observed RH
trends.
Although oceanic evaporation is decisive on moisture supply to continental regions
(Gimeno et al., 2010), several processes, which are not considered in this study, may
strongly affect RH and precipitation in continental areas. Under a generalized increase
of SST at the global scale (e.g. Rayner et al., 2003; Deser et al., 2010), a higher
moisture supply to the atmosphere and a strong decrease in RH can be expected in
different regions. Some of these regions (e.g. Amazonia and Western North America)
show an average high moisture supply from oceanic areas. On one hand, a global
warming signal does not necessarily imply above-normal oceanic evaporation. . Here,
we indicated that oceanic evaporation trends for 1979-2014 showed strong spatial
variability at the global scale, with dominant positive trends. Nevertheless, large areas
also exhibited insignificant trends and even negative evaporation trends. While SST
increase is mainly associated with radiative processes, evaporation processes are mainly
controlled by a wide range of meteorological variables that impact the aerodynamic and
radiative components of the atmospheric evaporative demand (AED) rather than SST
alone (McVicar et al., 2012b). This is consistent with the finding that global mean
precipitation or evaporation does not scale with Clausius–Clapeyron (Held and Soden
2006). Due to the unlimited water availability over oceans, air vapor pressure deficit is
expected to be driven by the Clausius-Clapeyron relation. However, changes in solar
radiation and wind speed can also influence the evaporation evolution (Yu, 2007;
Kanemaru and Masunaga, 2013). As such, given the slow oceanic evaporation trends in
large regions of the world, RH trends in some of the analyzed regions could be seen in
the context of a lower water supply to maintain RH constant, particularly with air
temperature rise.
Herein, we have not considered possible changes in other variables that could explain
the low relationship between RH, oceanic evaporation and oceanic moisture
contribution. Among these variables, we have not considered –for example- the role of
the "effectivity" of the oceanic moisture (Gimeno et al., 2012). This variables is of
particular importance, because oceanic evaporation might not reach the target region,
due to some geographical constraints (e.g. topography). Another relevant variable is the
transport mechanisms between the source and target regions, which  is a key variable in
some regions like Western North America (Wei et al., 2016b). Moreover, moisture
source regions are not stationary, as the intensity of humidity can vary greatly from one
year to another (Gimeno et al., 2013). This aspect could be another source of
uncertainty in the explanatory factors of current RH trends. Furthermore, other different
factors that control atmospheric humidity and RH have not been approached in this
study. Sherwood (1996) indicated that RH distributions are strongly controlled by
dynamical fields rather than local air temperatures. This suggests that atmospheric
circulation processes could largely affect the temporal variability and trends of RH. A
range of studies indicates noticeable changes in RH, in response to low-frequency
atmospheric oscillations, such as the Atlantic Multidecadal Oscillation (AMO) and El
Niño-Southern Oscillation (e.g. McCarthy and Toumi, 2004; Zhang et al., 2013),
regional circulation (Wei et al., 2016a and 2016b), as well as changes in the Hadley Cell
(HC) (Hu and Fu, 2007). Wright et al. (2010) employed a global climate model under
double $CO_2$ concentrations to show that tropical and subtropical RH is largely
dependent on a poleward expansion of the Hadley cell: a deepening of the height of
convective detrainment, a poleward shift of the extratropical jets, and an increase in the
height of the tropopause. Also, Laua and Kim (2015) assessed changes in the HC under
$CO_2$ warming from the Coupled Model Intercomparison Project Phase-5 (CMIP5 model
projections. They suggest that strengthening of the HC induces atmospheric moisture
divergence and reduces tropospheric RH in the tropics and subtropics. This spatial
pattern resembles the main areas showing negative trends in RH in Northern as well as
Southern hemisphere.

**5. Conclusions**
This study analysed relative humidity (RH) trends at the global scales using
observations and ERA-40 data. It extended further to link RH trends with a range of
variables, which can give indications on the possible oceanic and continental
contribution to RH trends. As opposed to the widely-accepted constant RH scenario
under global warming, our results suggest significant RH trends over many regions
worldwide, but including: There are positive trends in RH over specific regions (e.g.
high latitudes of the North Hemisphere, Northern South America, India, West Sahel),
which is in contrast to the generally dominant negative trend at the global scale. This
decrease is mostly linked to the temporal evolution of RH during the boreal warm
season.
There is a strong diversity in the observed RH trends, highlighting the complex and
divergent role of different physical processes and drivers, including dynamic and
thermodynamic processes. In general the supply of specific humidity is a main source of
the observed RH trends since there is a high agreement between RH and specific
humidity trends at the global scale, suggesting that moisture deficit contributes to RH
variability, in opposition to atmospheric warming. This finding suggests that the
evolution of specific humidity in vast areas of the world has not provided the necessary
humidity to maintain RH constant according to the observed warming trends. This
feature is important, given its implications in terms of atmospheric evaporative demand
and aridity conditions under the current climate change scenario.
This study also analyzed the possible contribution of continental and oceanic moisture
supply to explaining the magnitude and spatial patterns of RH trends. For this purpose,
fourteen regions were defined and the contribution of continental and oceanic sources to
RH in these regions was assessed using a Lagrangian scheme. Results indicate that no
single physical mechanism can be responsible for the observed trends in RH at the
global scale. Globally, there are two well-recognized hypotheses for explaining the
possible decrease in RH under a global warming scenario: (i) the land water supply by
means evapotranspiration processes, and (ii) the insufficient oceanic moisture supply to
maintain continental RH constant under different warming rates between continental
and oceanic regions. Our findings stress that these two hypotheses could act together to
explain recent RH trends. However, although it is quite difficult to establish a direct
causality between RH and different underlying processes and driven variables using
different empirical sources, the observed decrease in RH in some regions (e.g. La Plata)
can be linked to the lower water supply from land evapotranspiration. In other regions,
the empirical relationships suggest dynamic and thermodynamic mechanisms related to
moisture supply from oceanic source regions (e.g. Amazonia and Western North
America). Taken together, these physical mechanisms could coexist in some analyzed
regions, given the strong relationship found between precipitation, RH and land
evapotranspiration. This strong coupling among these variables makes it difficult to
establish a direct physical attribution of RH variability.
Overall, this study confirms the strong complexity of determining a general physical
process that may explain the complex spatial patterns of RH trends, particularly at the
global scale. As such, further research is still needed to unravel the complex physical
factors driving the dominant RH negative trends over large continental regions. The
availability of long-term historical and reanalysis data and the advancement of
modelling approaches is an asset in any future research to explore whether the land and
oceanic processes drive the observed RH trends
Understanding current RH is relevant in hydroclimatic research, due to its impacts on
atmospheric evaporative demand, crop development and yield, forest fire risk,
bioclimatic comfort, besides other hydrological processes. This study provides the first
comprehensive analysis of RH at the global scale based on empirical information,
comprising state-of-the art modelling approaches and forcing scenarios.

**Acknowledgements**
We thank Dr. H. F. Goessling and the two anonymous reviewers for their insightful and
constructive comments, which improved significantly the quality of the manuscript.
This work was supported by the EPhysLab (UVIGO-CSIC Associated Unit), PCIN-
2015-220, CGL2014-52135-C03-01, CGL2014-60849-JIN, *Red de variabilidad y*
*cambio climático* RECLIM (CGL2014-517221-REDT) financed by the Spanish
Commission of Science and Technology and FEDER, IMDROFLOOD financed by the
Water Works 2014 co-funded call of the European Commission and INDECIS, which is
part of ERA4CS, an ERA-NET initiated by JPI Climate, and funded by FORMAS (SE),
DLR (DE), BMWFW (AT), IFD (DK), MINECO (ES), ANR (FR) with co-funding by
the European Union (Grant 690462).

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

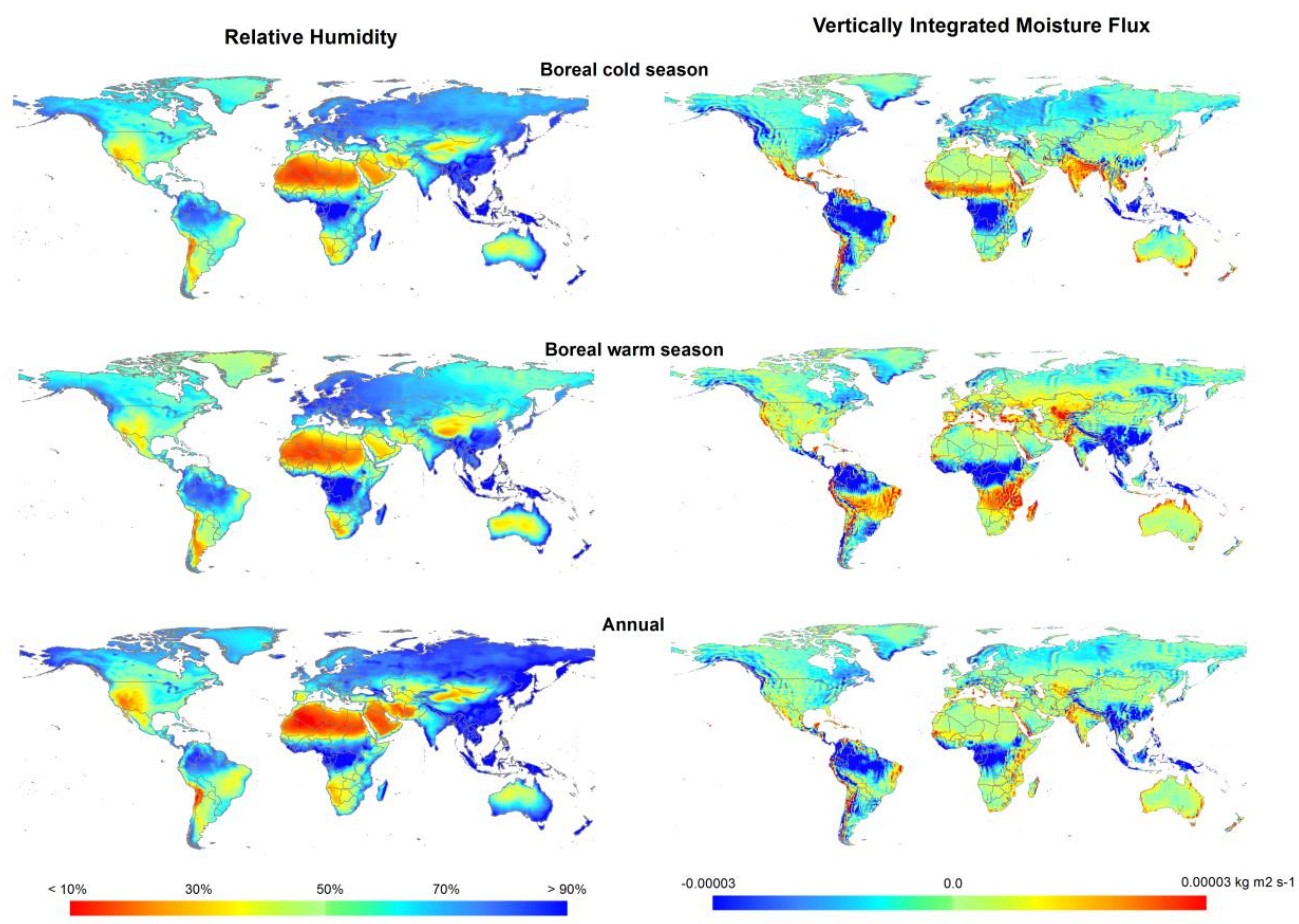

Figure 1. Annual and seasonal averages of RH and Vertically Integrated Moisture Flux (VIMF) based on ERA-Interim dataset.

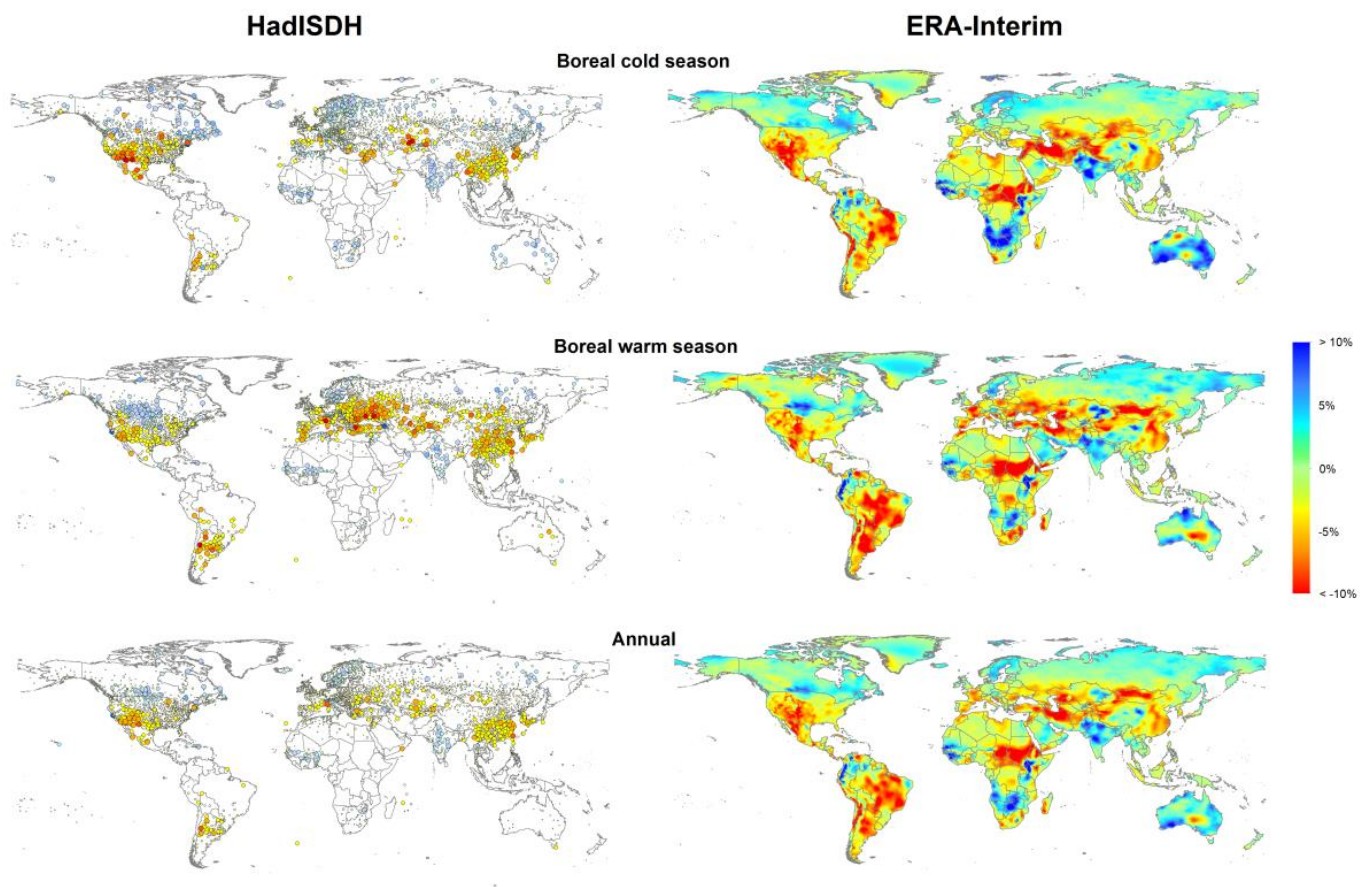

Figure 2. Spatial distribution of the magnitude of change of RH (% per decade) over the period 1979-2014 from HadISDH (left) and ERA-Interim dataset (right). Results are provided for the boreal cold (October-March) and warm (April-September) seasons and annually.

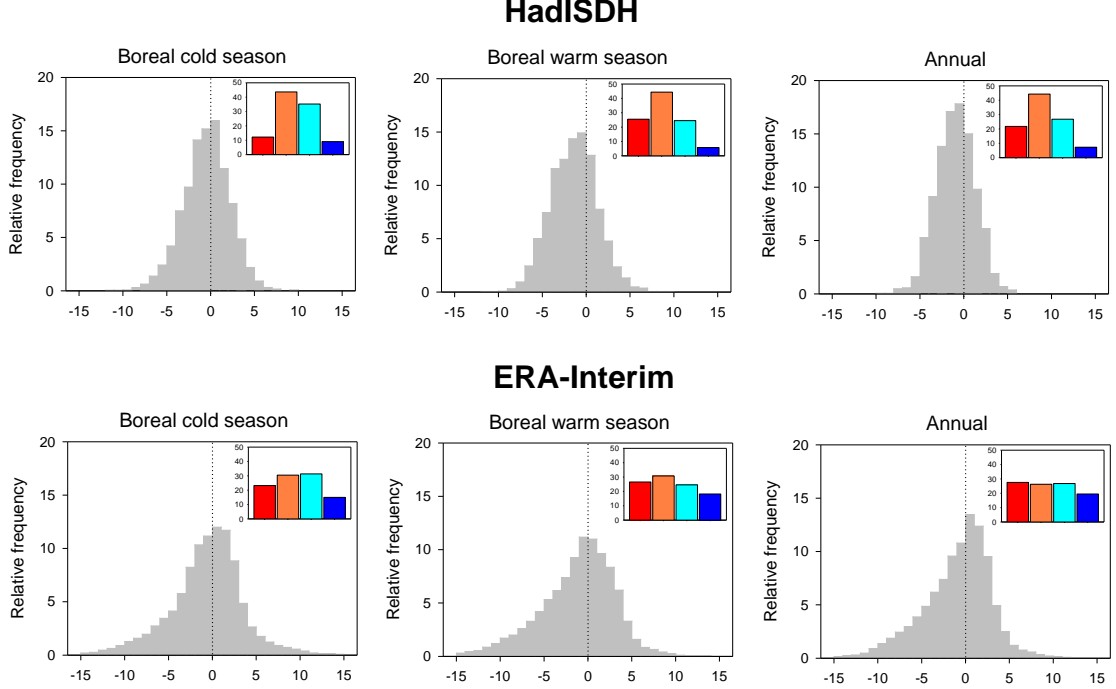

Figure 3: Relative frequencies (%) of the RH magnitude of change in the HadISDH and ERA-Interim datasets. Color bar plots represent the percentage of stations (from HadISDH) and world regions (from ERA-Interim) with positive and significant (p < 0.05) trends (blue), positive insignificant trends (cyan), negative insignificant trends (orange) and negative and significant trends (red).

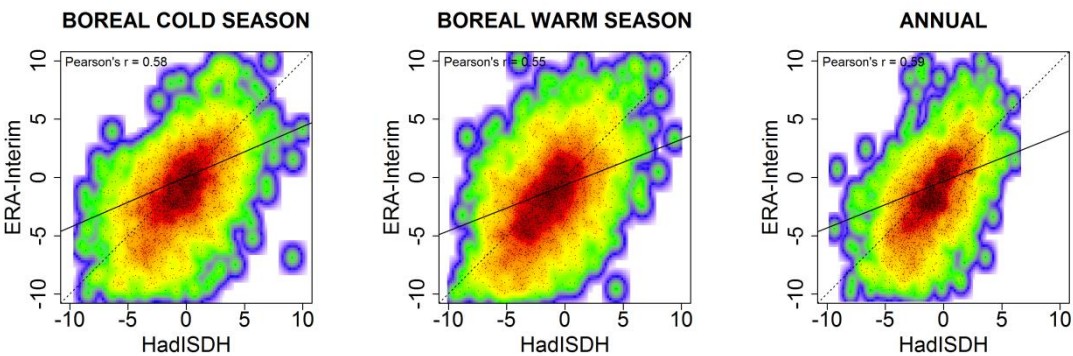

Figure 4: Scatterplots showing the global relationship between the magnitude of change in RH with HadISDH stations and ERA-Interim dataset at the seasonal and annual scales. Colors represent the density of points, with red color showing the highest density of points.

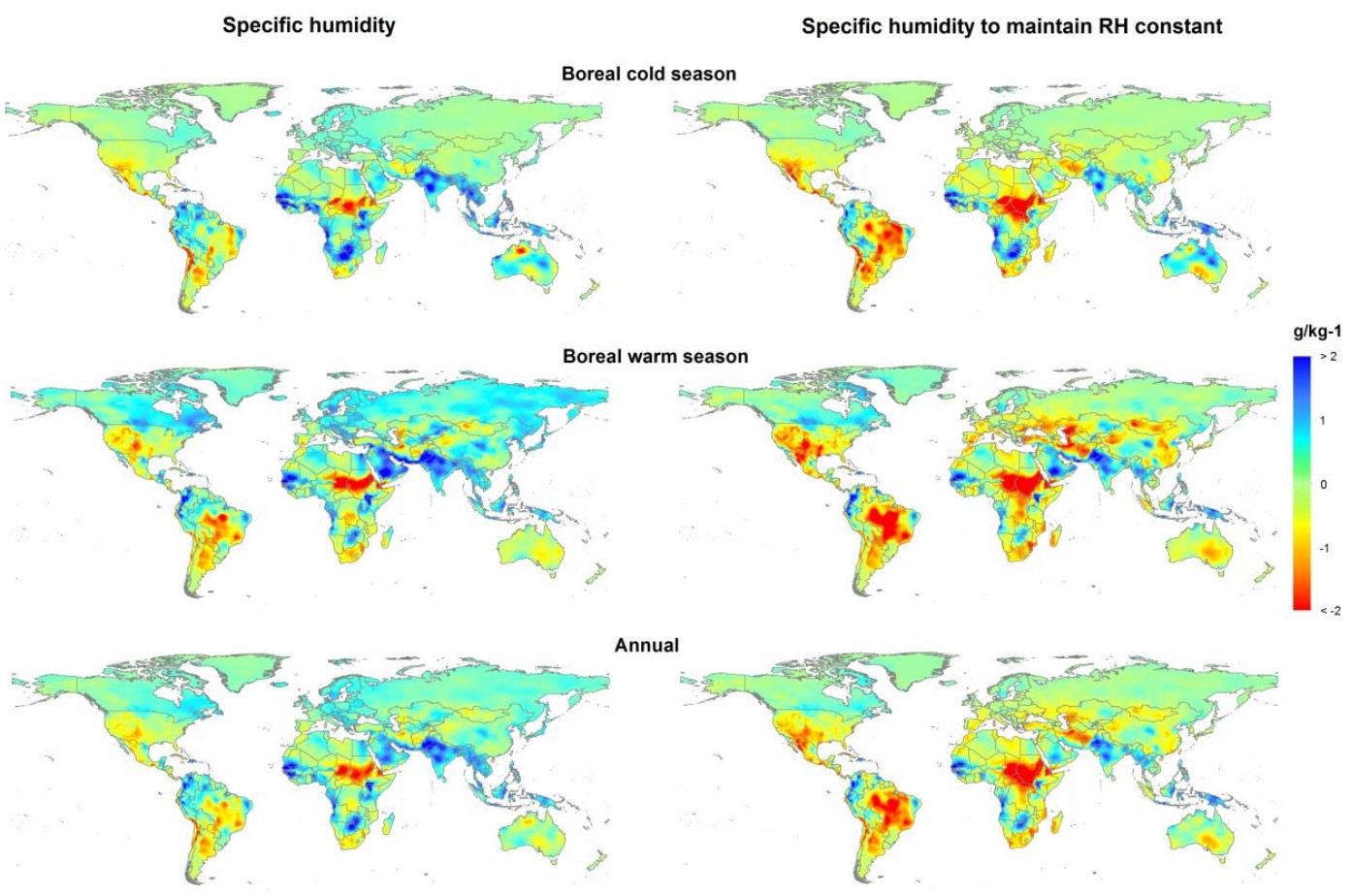

Figure 5: Spatial distribution of the seasonal and annual magnitudes of change in specific humidity (g/kg$^{-1}$) (left) and the deficit/surplus of specific humidity to maintain the RH constant with the levels of 1979 according to the land air temperature evolution (from the CRU TS v.3.23 dataset) for 1979-2014.

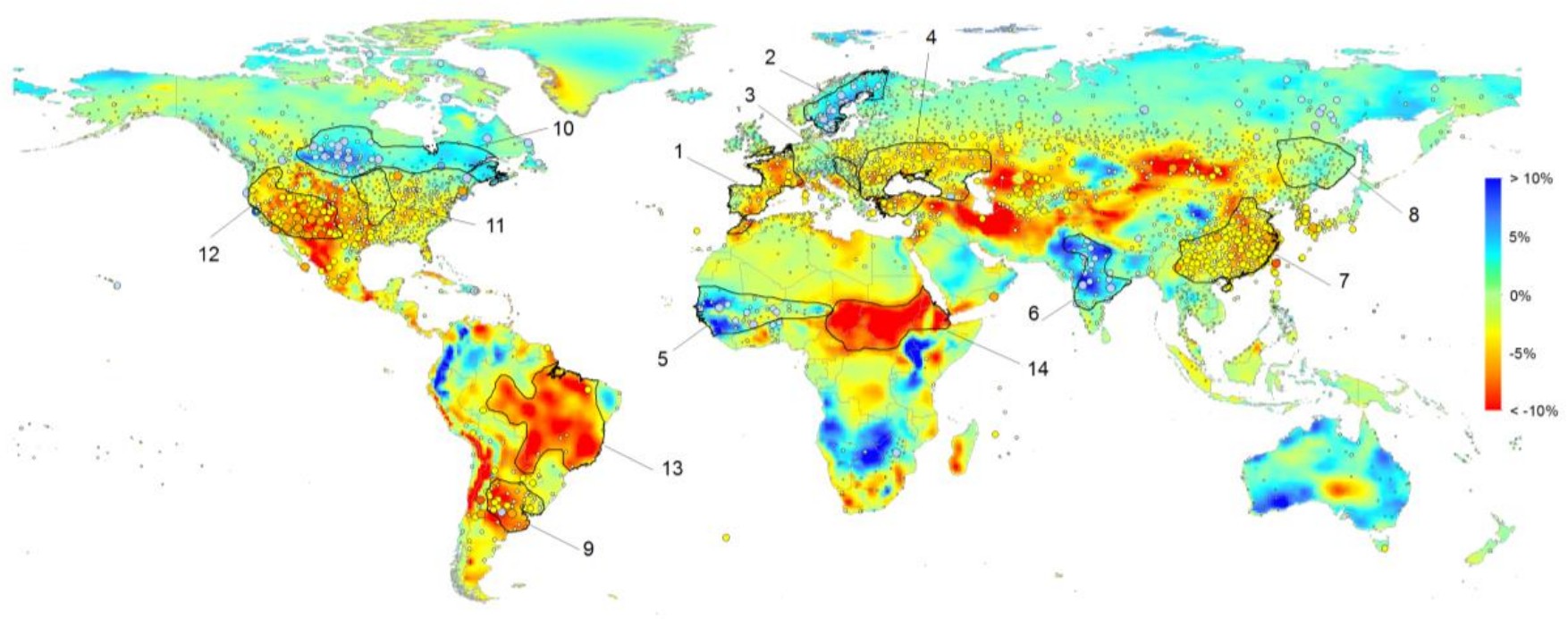

Figure 6: Distribution of the 14 world regions, with high consistency in RH trends between the HadISDH and the ERA-Interim datasets. These regions were selected for the identification of the oceanic and land humidity sources by means of the FLEXPART scheme.

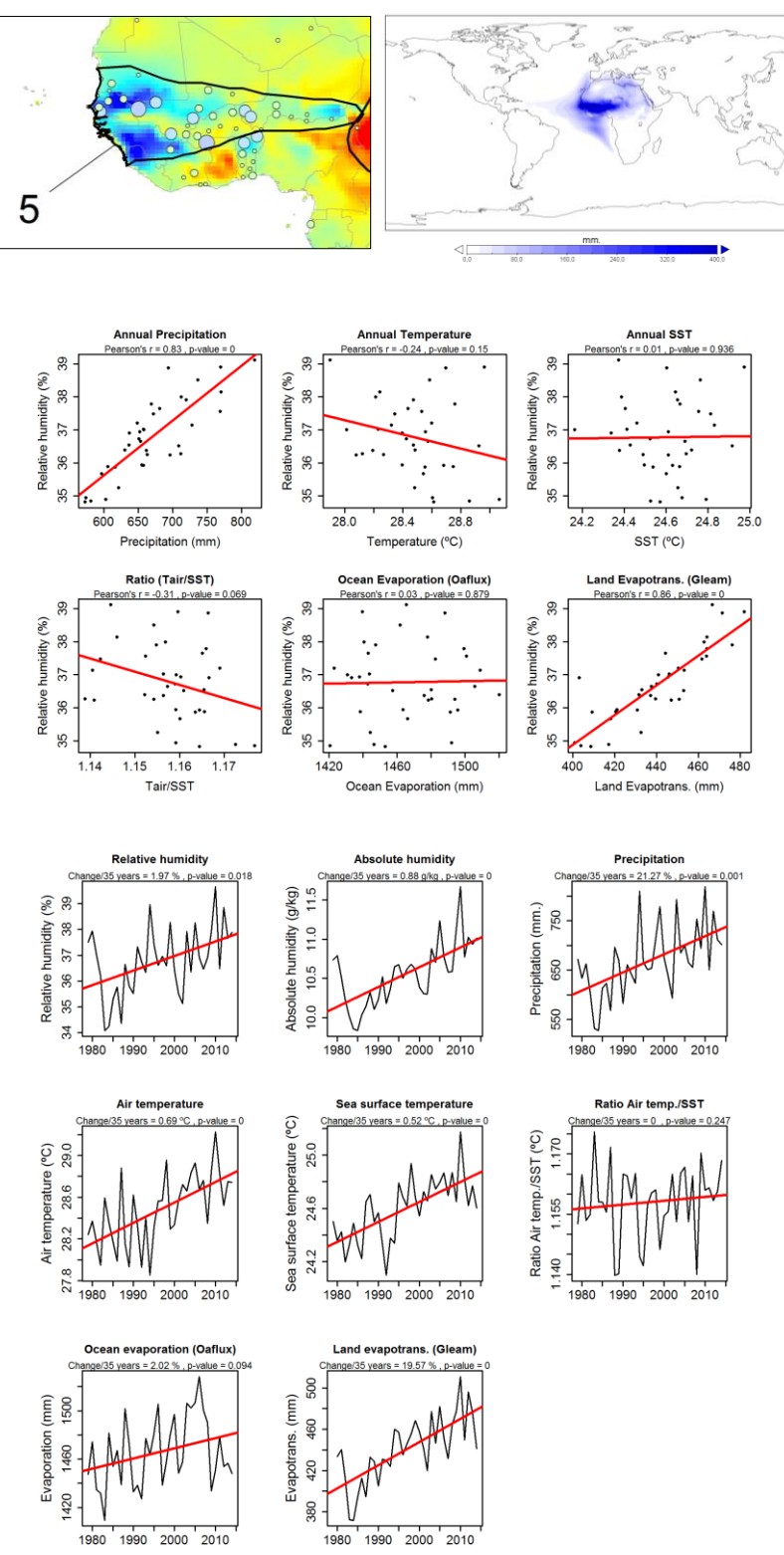

Figure 7: Top left: Annual RH humidity trends in the West Sahel (region 6), Top right: average (E-P)>0 at the annual scale to identify the main humidity sources in the region (mm year$^{-1}$). Center: Relationship between the de-trended annual RH and the de-trended annual variables for 1979-2014. Bottom: Annual evolution of the different variables corresponding to the West Sahel region. The magnitude of change and significance of the trend is indicated for each variable.

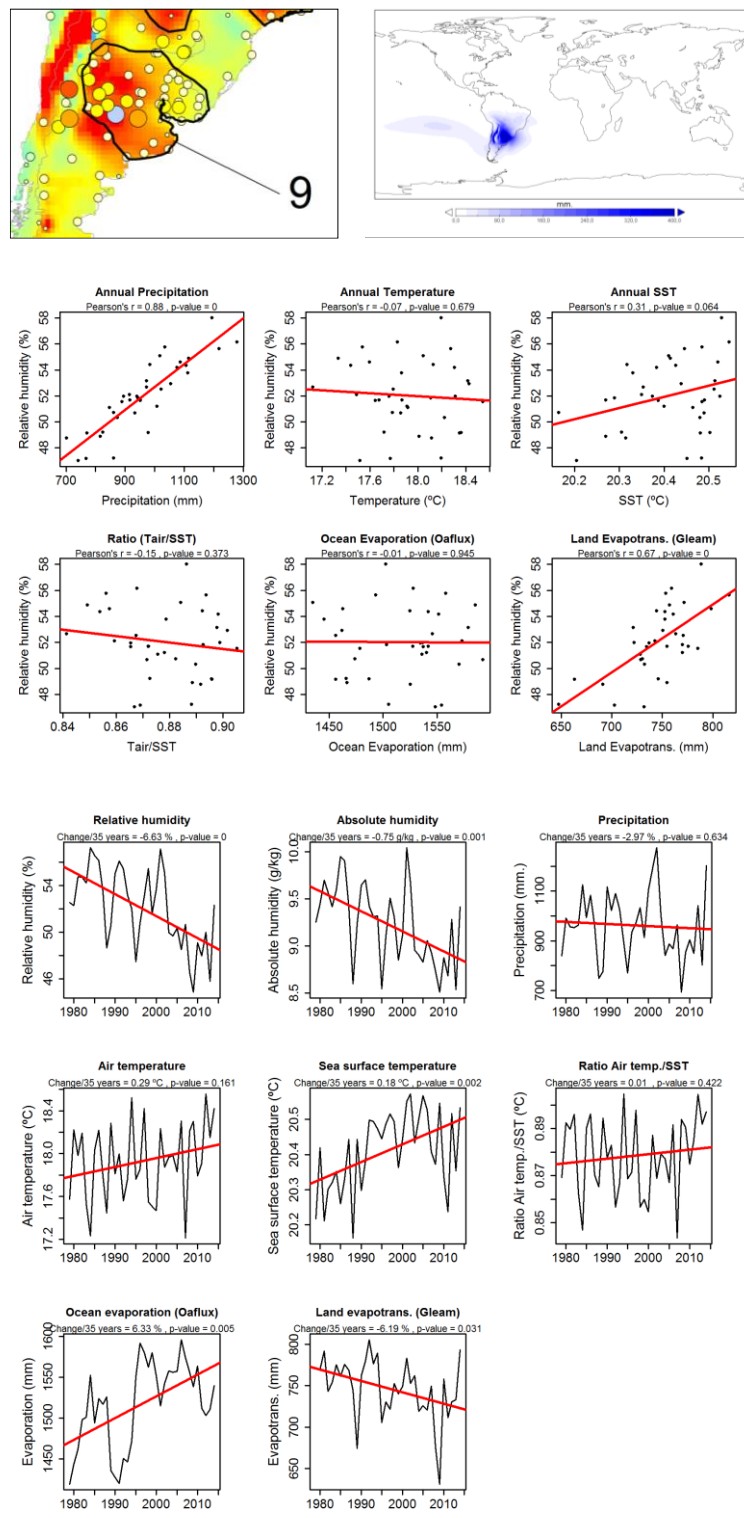

Figure 8: The same as Fig. 7 but for La Plata (region 9).

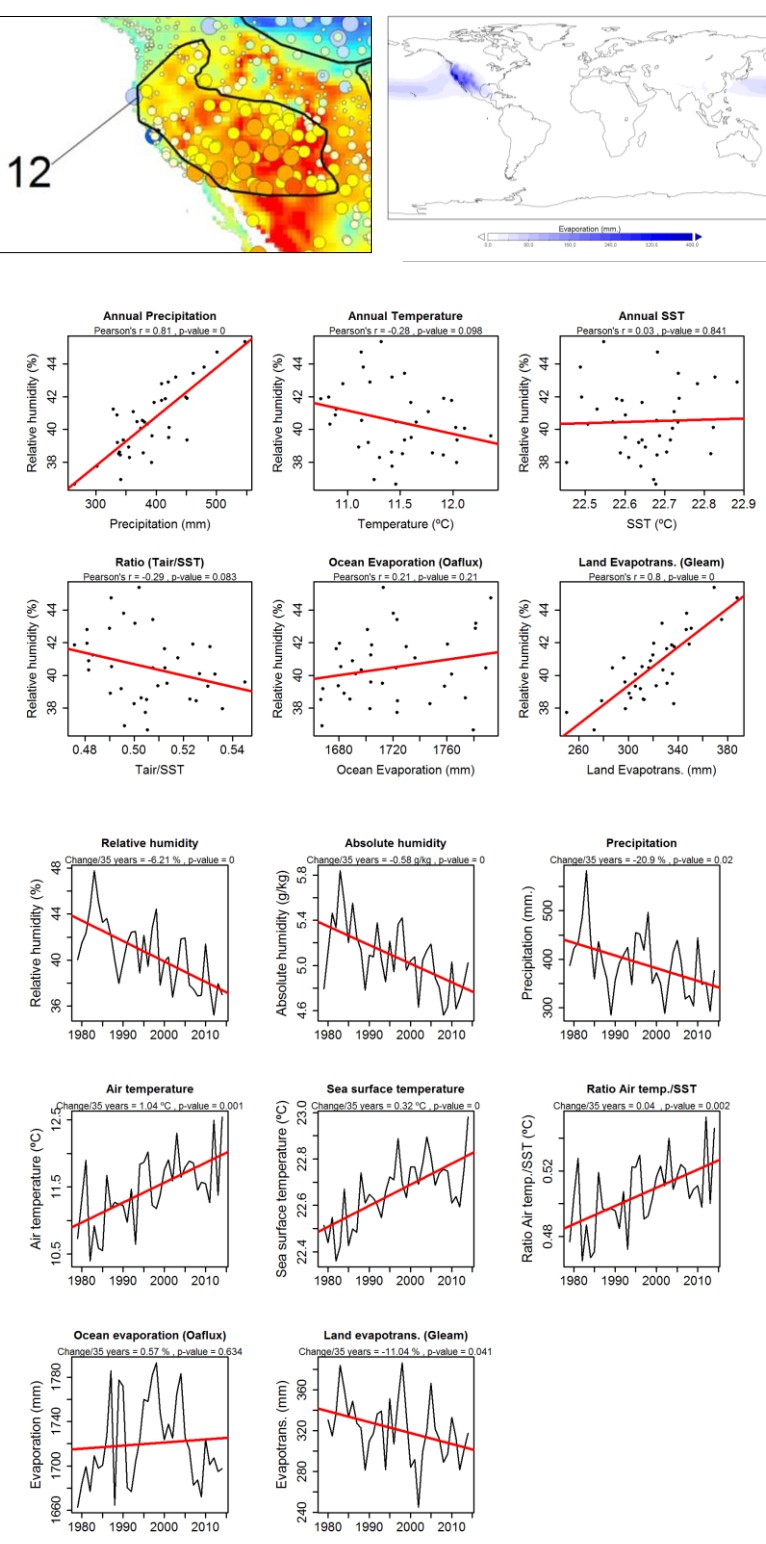

Figure 9: The same as Fig. 7, but for West North America (region 12).

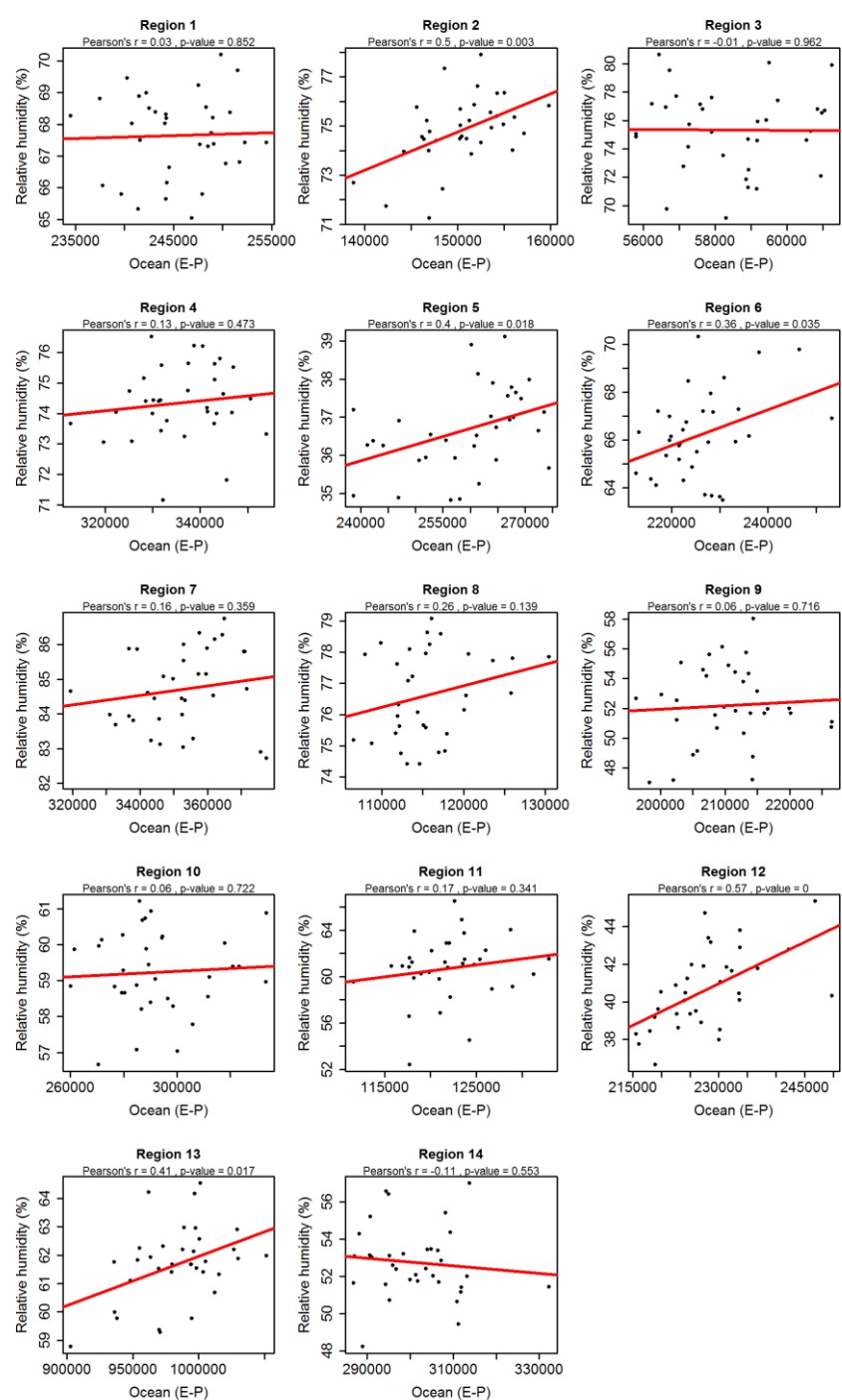

Figure 10: Relationship between the annual ocean contribution to annual precipitation (E-P) and the annual RH in the target regions.

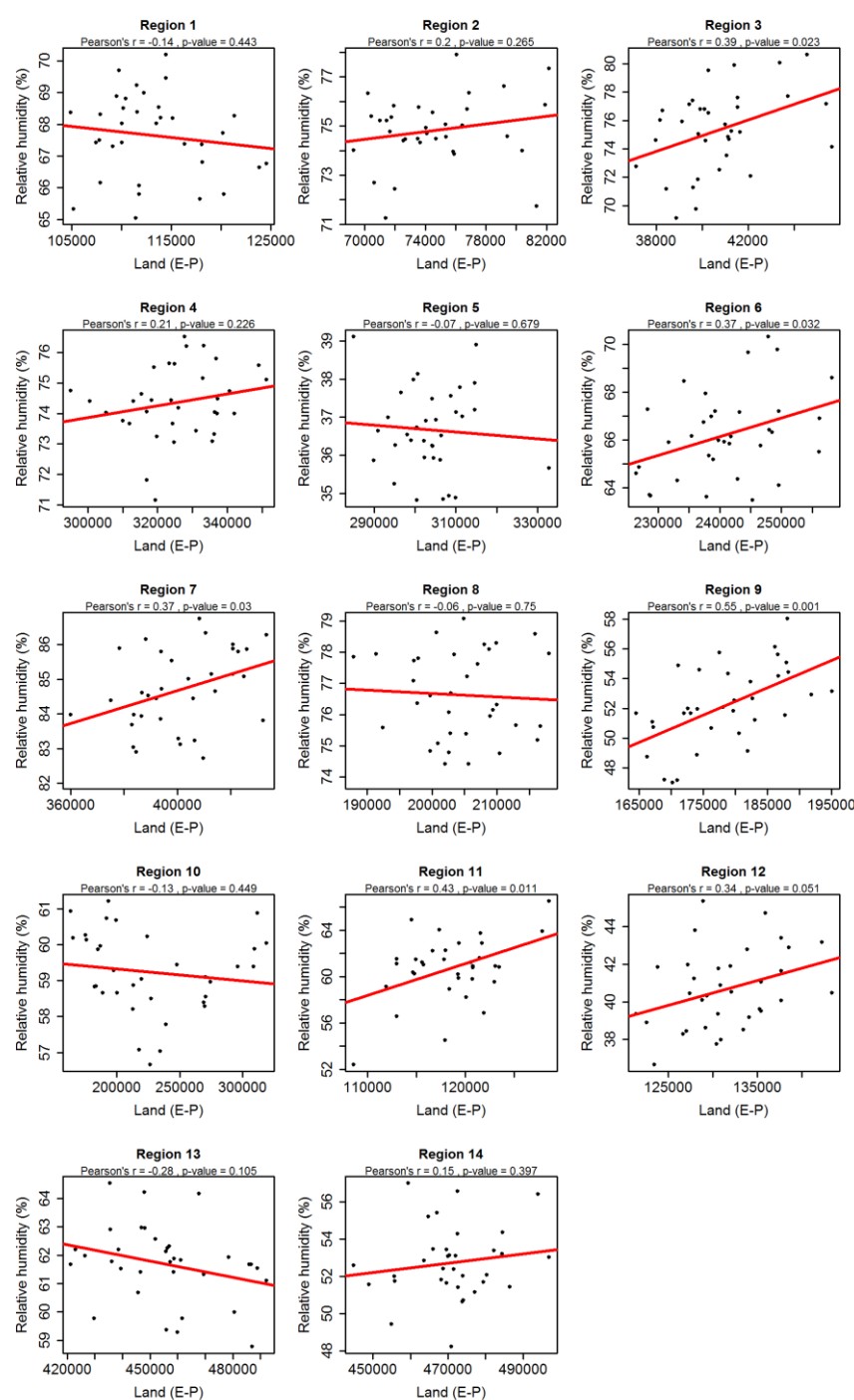

Figure 11: Relationship between the annual land contribution to annual precipitation (E-P) and the annual RH in the target regions.

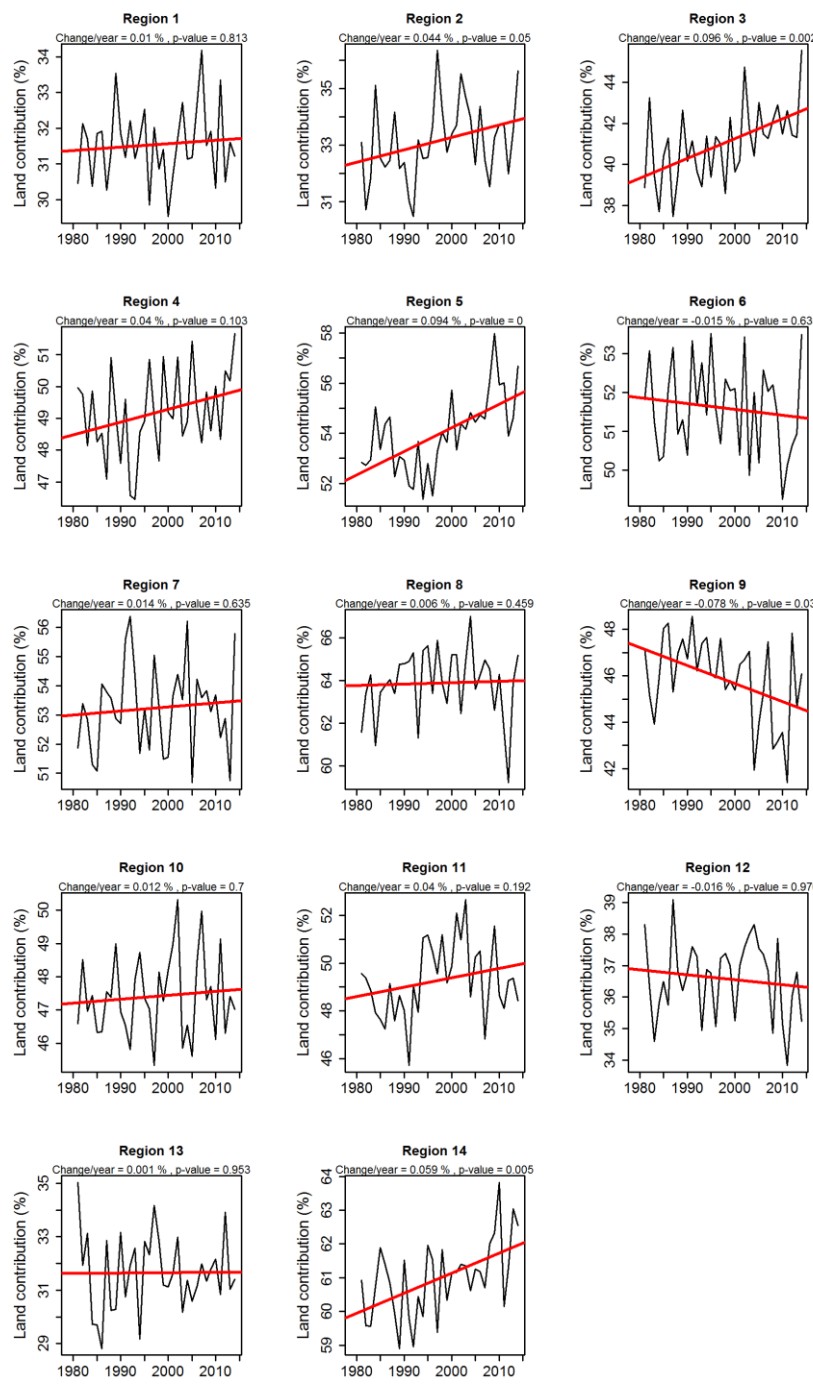

Figure 12: Evolution of the land contribution (%) to annual precipitation in the different target regions

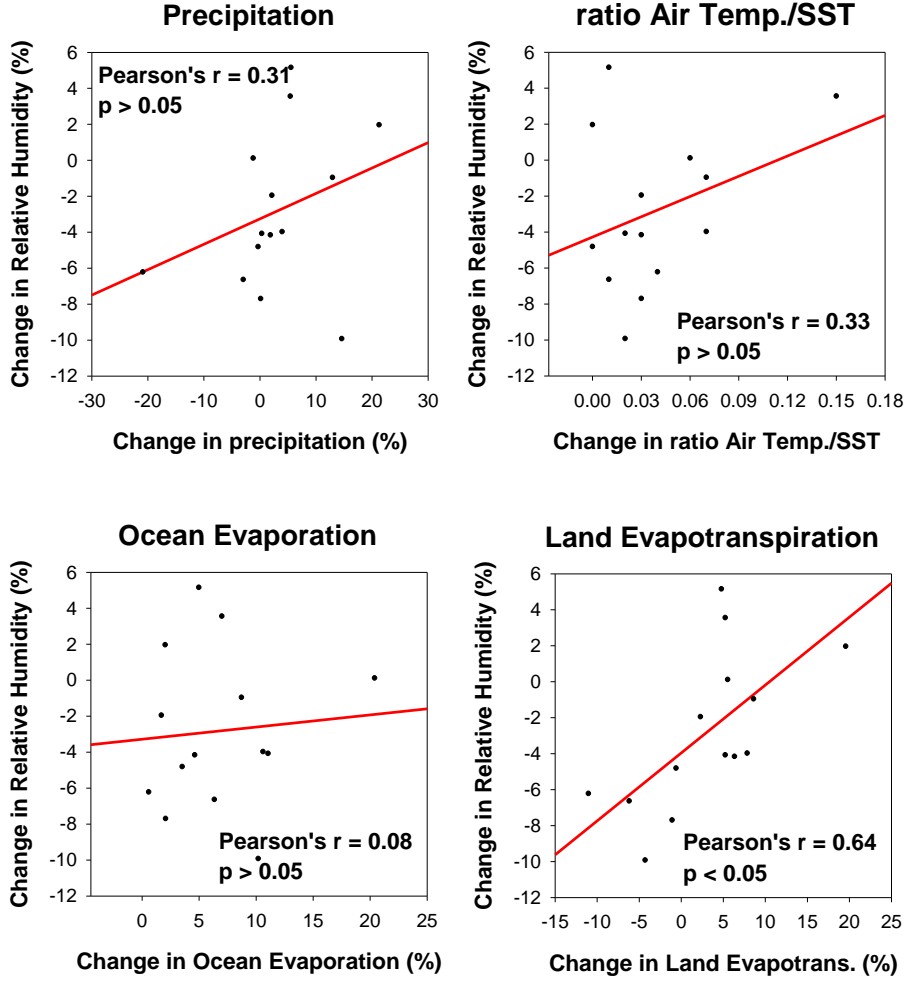

Figure 13: Relationship between the average annual magnitude of change in RH identified in each one of the 14 analyzed regions and the annual magnitude of change in precipitation, the ratio between air temperature/SST, oceanic evaporation and land evapotranspiration.

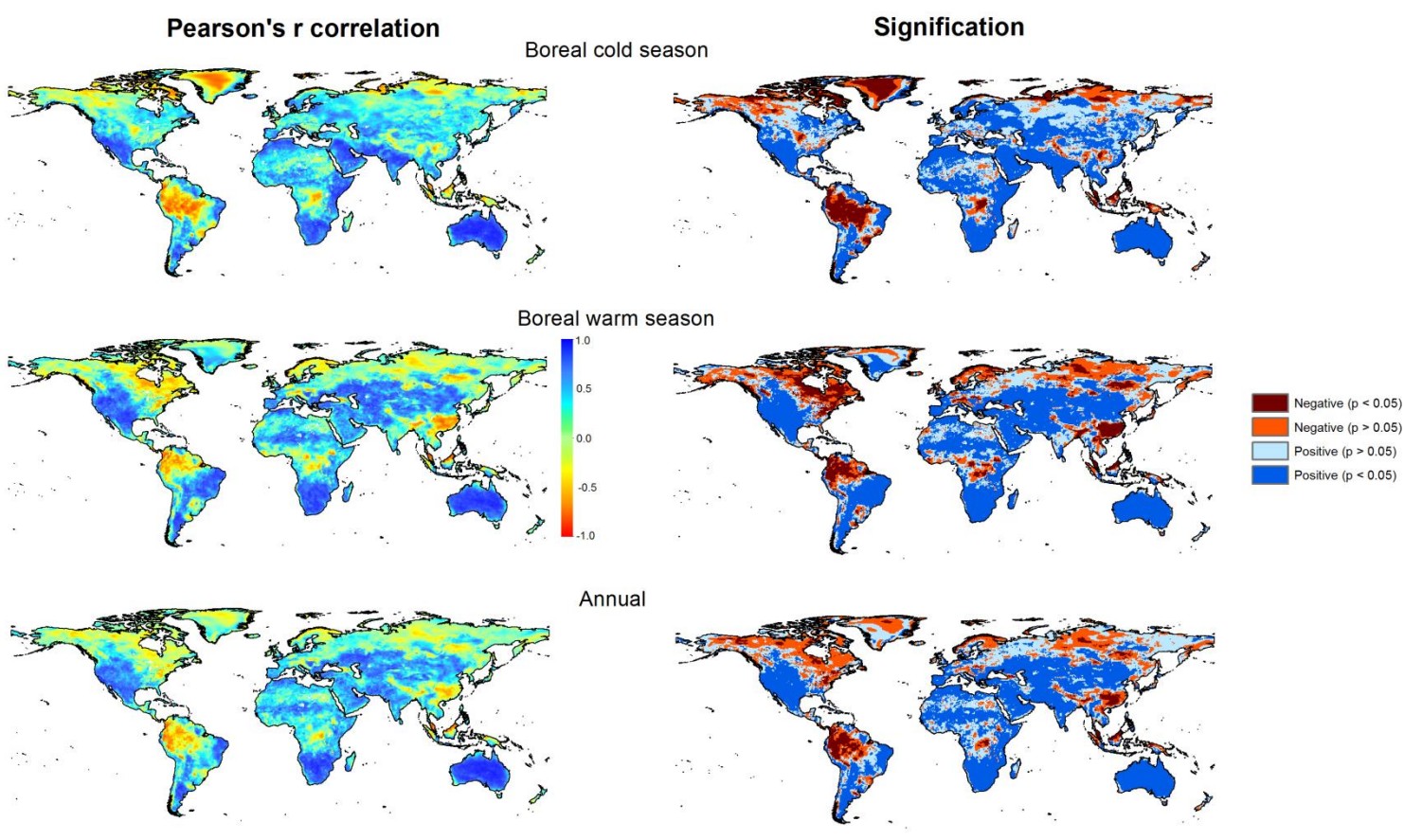

Figure 14: Spatial distribution of the Pearson's r correlations between the detrended RH and land evapotranspiration series at the annual and seasonal time scales. The statistical significance of the correlations is also shown.

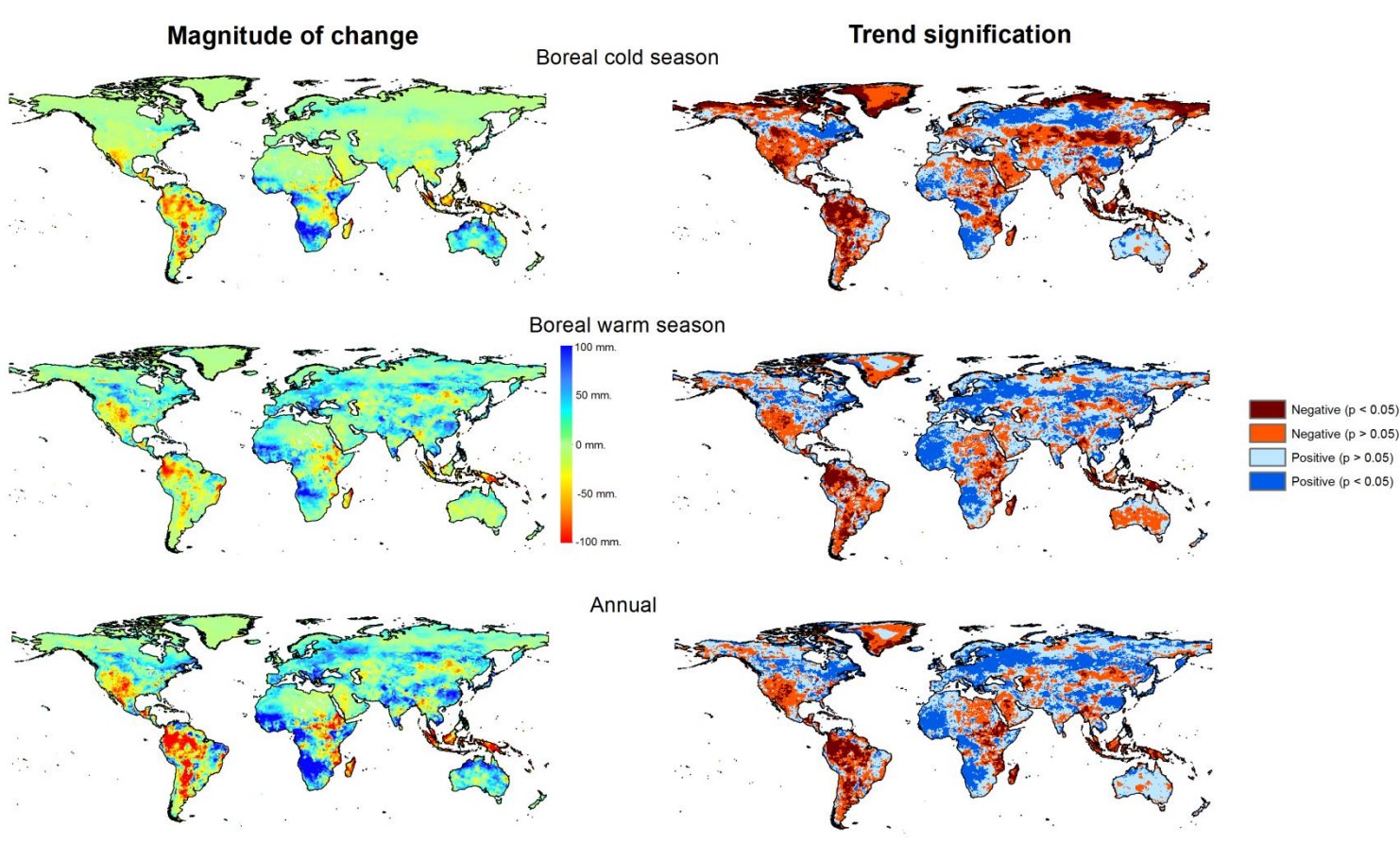

Figure 15: Spatial distribution of the magnitude of change in the annual and seasonal land evapotranspiration (1979-2014) and statistical significance of trends.

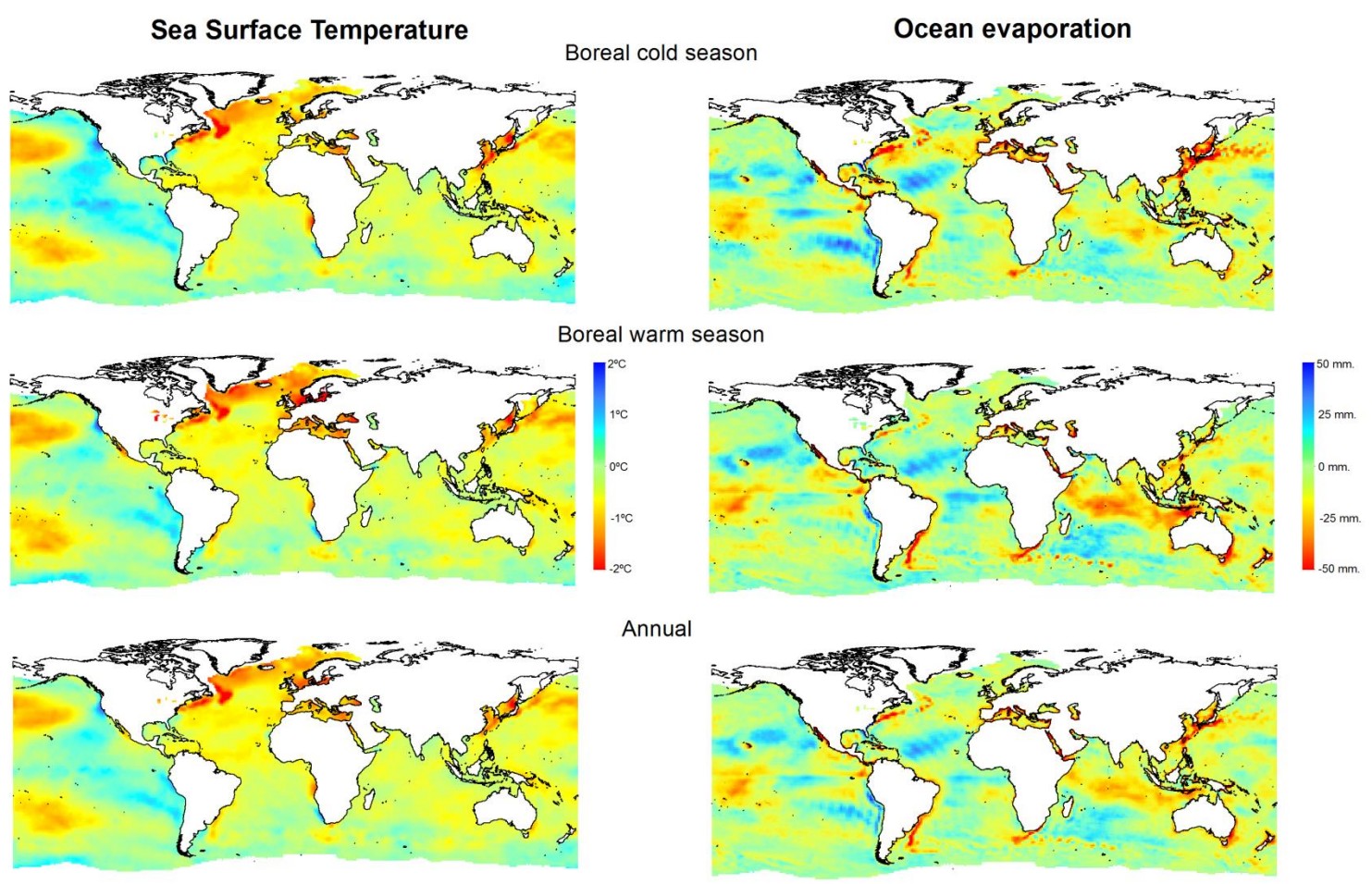

Figure 16: Annual and seasonal magnitude of change of SST and OAFLUX oceanic evaporation for 1979-2014.