# Peer review of "Recent changes of relative humidity: regional connection with land and ocean processes"

_Earth System Dynamics, 2017_

## Referee Comment (RC1) · Anonymous Referee #1 · 11 Jul 2017

The paper analyses changes in surface relative humidity (RH) in the last 35 years. It is mostly based on ERA-Interim reanalysis data, but observations (HadISDH) are also considered. The main technique used is a Lagrangian analysis of the moisture source regions.

I think the topic is very important, and in scope for Earth System Dynamics. The paper will be a valuable contribution to the field, but in my opinion it requires some revisions.

Major Issues

1. My main criticism is that the paper lacks clear conclusions. Section "Discussion and Conclusions" is dominated by a discussion of other relevant studies, with too little

effort to distill out what is new. My suggestion is to separate out "Conclusions" to a separate section. There, clearly state what is new, what are the specific conclusions of this study.

2. l139-l151: Text and formulas for RH computation:

a. Why is the decision, whether the ice or liquid equilibrium pressure is used, based on the wet bulb temperature (and not, for example, the physical temperature)? Please justify or change.

b. There are two different equilibrium pressures to consider, one for liquid water and one for ice. Authors chose to use the ice one for temperatures below 0°C. There is no direct physical reason for this, since there will be large open water areas even at sub-zero air temperature and vice versa. But I guess it is a reasonable first choice if information on the surface itself is not available. However, the authors should be clear about that it is a choice they are making, and not dictated from physics.

c. Generally, I find text and forumlas here confusing and outdated. From Eq. 1 it follows simply that

$RH = 100*e/es(T) = 100*es(Td)/es(T)$

where RH = relative humidity in percent, e = water vapor partial pressure, T = physical temperature, Td = dewpoint temperature, es() = equilibrium water vapor pressure.

So, to calculate RH from Td, all that is needed is a valid parameterisation of es(T). An accepted modern one is given in Murphy, D. M. and T. Koop (2005), Review of the vapour pressures of ice and supercooled water for atmospheric applications, Q. J. R. Meteorol. Soc., 131(608), doi:10.1256/qj.04.94. (They give two different es(T) parameterisations, one for liquid, and one for ice.)

3. I would like to see an overview figure of the RH mean state and the E-P mean state, before the trends are discussed. Perhaps in the same style as Figure 1, for cold season, warm season, and anual mean. And also a short discussion, along with the

figure. This is important to put the changes into perspective. Also, in my understanding the "null hypothesis", based on simple thermodynamic arguments and the simplistic assumption that the circulation does not change, is that the existing E-P pattern is enhanced under global warming. (The "dry gets drier, wet gets wetter paradigm (Held, Isaac M. and Brian J. Soden (2006), Robust Responses of the Hydrological Cycle to Global Warming, J. Climate, 19(21), 5686-5699, doi:10.1175/JCLI3990.1.).)

Minor Issues

The paper contains some English errors and idiosyncrasies. Example: "On the contrary", used in several places, where I think the authors mean "on the other hand". I recommend a careful proof-reading, preferably by a native speaker.

l47 "water holding capacity": Please replace by "equilibrium amount of water vapor". (Holding capacity is physically wrong, since the air does not "hold" the water vapor in any way. The CC equation describes the equilibrium pressure of water vapor with liquid water.)

l49 "could increase": Please replace by "is expected to increase".

l83 "there are unavailable studies": Rearrange: "studies...are unavailable"

l103 "challengeable" –> "challenging"

l239 "moisture support": Perhaps replace by "moisture supply"? (Meaning of support is not clear here.)

l277: "positive (E-P) field": I'm confused. Isn't Figure 1 showing just the RH trends? How does the E-P field enter the figure?

l364 "controlled by": I would say "correlated to". How do you know what controls what?

l378 [RH has increased] "as a consequence of changes in the continental humidity sources": Why have they increased?

l407 "air temperature and SST ratio" –> "and air temperature to SST ratio"

l492 "Thus, although some regions showed positive changes in the oceanic evaporation, the amount of increase was much lower than that found for SST, suggesting a general positive trend in most of the world's oceans (Supplementary Figure 48, Supplementary Table 1)."

Confusing. Did you mean:

"Thus, although some regions showed positive changes in the oceanic evaporation, the amount of increase was much lower than that found for SST, which suggests that SST changes do not drive evaporation changes (Supplementary Figure 48, Supplementary Table 1)."

l591 "This finding indicates that while different model experiments fully supported the hypothesis that the different warming rates between oceanic and continental areas can explain the projected decrease in RH under climate change conditions, our results for 14 different regions in the world are contradictory, given that most of these regions exhibited a negative RH trend for 1979- 2014."

What is the contradiction? The differential land sea warming mechanism predicts a decrease of RH over land, and you find a negative RH trend. I'm not sure if this is a language issue, or if there is a fundamental point that I'm missing.

l638 "Hadley Cell" –> "Hadley Cell (HC)"

Figure 2: Please add a vertical zero line, so that one can judge whether trends are positive of negative.

Figures 6-8: Subplots are too small, titles almost impossible to read for me.

Figures 10-11: "Signification" –> "Significance"

---

## Referee Comment (RC2) · H. F. Goessling (Referee) · 22 Sep 2017

**Summary**

The authors analyse changes in relative humidity (RH) over continental regions during the period 1979-2014. For a number of regions with clear positive or negative trends in RH, they relate the RH changes to possible trends in local air temperature and local precipitation as well as continental evapotranspiration (ET), SST, and ocean evaporation. The authors determine relevant source regions for which to compute the latter three quantities based on an existing Lagrangian vapour tracking algorithm. They analyse relationships between these quantities and RH not only with respect to trends,

but also with respect to interannual variations. Generally, the clearest relations they find are positive correlations of RH with precipitation and ET, and they conclude that continental ET plays an important role as driver of continental RH changes.

The results shown in the paper are interesting and well presented (although the text needs quite a number of corrections in terms of language/grammar), and I think they merit publication. However, I am not convinced that the causality put forward in the interpretation is sufficiently evidenced, so I would recommend to phrase some conclusions more cautiously. Specifically, I would argue that the positive correlation between RH and continental ET does not prove the suggested causality. There are a number of other points, of which some are minor but some are not so minor, detailed below, that deserve clarification and/or revision.

Overall, I recommend the manuscript should be reconsidered after major revisions.

**Specific comments**

P1L20-22: "The aim was to account for the possible role of changes in air temperature over land, in comparison to sea surface temperature (SST), on RH variability." - This sentence seems to suggest that this was the only aim of the paper, but the role of land and ocean evapo(transpi)ration changes is obvisouly also accounted for ...

P1L22-25: "Results demonstrate a strong agreement between the interannual variability of RH and the interannual variability of precipitation and land evapotranspiration in regions with continentally-originated humidity." - After having read the paper, I wonder if the last part of this sentence is supported by the content: It appears that the authors have not systematically assessed how the fraction of "continentally-originated humidity" is related to the strength of this agreement. However, this would be quite interesting. I suggest to compute the fraction of continentally-originated humidity (sometimes called the continental recycling ratio) for each of the regions (and season), and to relate this fraction to the strength of the agreement to verify (or not) this statement.

[Figure]

P3L83: "Nonetheless, there are unavailable empirical studies that support ..." - In my view it would be more logical (and elegant) to rephrase this to something like the following: "However, there are no empirical studies available that support ...". (It seems that the term "nonetheless" is used in the sense of "however", which I think is not correct, also elsewhere in the paper.)

P3L84-86: "One of these hypotheses is related to the slower warming of oceans in comparison to continental areas. In particular, specific humidity of air advected from oceans to continents increases more slowly than saturation specific humidity over land. This would decrease RH over continental areas [...]. The observed decrease in RH over some coastal areas, which are adjacent to their sources of moisture, adds further uncertainty to this hypothesis." - Is the observed decrease in RH, be it over coastal areas or further inland, not rather supporting the hypothesis?

P4L87-96: As explained here, the land-atmosphere feedback seems to be "only" a positive feedback rather than one that could explain why RH is altered under global warming in the first place, no?

P4L88-96 and P4L100-103: In this context (continental recycling ratios, evaporation-precipitation feedback, and the role of circulation changes), I can not resist to recommend that our paper on exactly these aspects (Goessling and Reick 2011), where we have combined moisture tracing with an idealised perturbation experiment in a climate model, revealing that circulation changes play a major role, should also be considered; see also other related comments below.

P4L107: "advections that were" -> "advection that was"

P5L125-128: Could the authors clarify how exactly the gap-filling was done?

Sect2.1: I suggest to use consistent headers for subsubsections 2.1.1-2.1.5, that is, either referring always to the dataset described in that section or to the variable(s).

P6Eq1-8: Most of the equations need some corrections with respect to units. Note that

T and P have units which implies that some of the constants used need to have the same units.

Sect2.1.5: While in the other sections the time period is always explicitly mentioned, that's not the case here; please clarify.

P8L190: "The statistical significance of the time series was tested at the 95% confidence interval" - I suggest it should be "trend" instead of "time series", and "level" instead of "interval".

P8L195: "with uneven number of stations" - I suggest this should be something like "with low station density".

P9L213-235: First, it would be good to clarify in what way the particles are distributed in the vertical initially: Does their vertical distribution correspond to the specific humidity profile? Second, I am wondering what explains the relatively short "optimal lifetimes" of 4-7 days found (as reported later in Sect. 3.2), and in particular what this implies. Given that even with the global-mean residence time of ~10 days only the closest (1-1/e)*100% of a (typical) source region is captured, it appears that only half or even less is captured on average with shorter backward tracking times. It appears that contributions from adjacent sources (in particular from nearby land) are thereby overestimated compared to more remote contributions (in particular ocean). I suggest to clarify this.

P9L226: "as better as" - bad grammar.

P9L232-233: "the optimal lifetime selected for each region was that fulfills the minimum absolute difference between" - bad grammar.

P10L253-254: "the ratio between air temperature in the target region and SST in the source region" - If I am not mistaken, this is a quantity that depends on the units used for temperature (Kelvin or Degrees Centigrade). Also, wouldn't it be more straightworward to use simply the temperature difference instead of the ratio?

P10L260: "signification" - Should be "significance", right? (Occurs many times throughout the manuscript.)

P11L263-267: "While a pixel-to-pixel comparison does not produce a reliable assessment of the possible contribution of land evapotranspiration to RH changes, given that the source of moisture can apparently be far from the target region, we still believe that this association can give insights on the global influence of land evapotranspiration on RH changes." - Here and generally, I have the impression that the suggested causality is not sufficiently attested and discussed. I would argue that increased land ET tends to be caused primarily by increased precipitation (except in very humid regions), and that anomalous precipitation (caused, e.g., by circulation anomalies) is simply accompanied by corresponding RH anomalies. And I think this is still largely valid for a non-local comparison, where land ET is determined for the "source region", because (i) the source region tends to overlap strongly with the target region and (ii) also most of the non-overlapping part is rather close, where spatial correlations of synoptic-scale anomalies are still high. I suggest this could be the main explanation for the positive correlations between RH, precipitation, and land ET.

P12L290: "uneven distribution" -> "low density"?

P12L302: "West Sahel" - Should be "East Sahel", right?

P13L315-317: "On the contrary, areas of complex topography in the Northern Hemisphere, Australia, India, Northern South America and Africa showed positive trends." - Can the authors comment why this should be so?

P13L324-326: "high consistency between the HadISDH and the ERA-Interim datasets in terms of both the magnitude and sign of change of in RH (Supplementary Figures 2 and 3)" - Are these figures not showing the correlation of interannual variations instead of the "magnitude and sign of change" (where the latter sounds as if the long-term trend is referred to)?

P13L328-P14L337: If I am not mistaken, according to Figure 4, q has decreased less

than it would have had to decrease in order to maintain RH constant (apparently due to a cooling trend?) in the mentioned regions (Southwest North America, the Amazonian region, Southern South America, and the (eastern!) Sahel). It appears that this corresponds to an INCREASE in RH in those regions, instead of the decrease shown in Fig. 1. Please clarify this contradiction.

Sect.3.2: I suggest to structure this subsection with subsubsections corresponding to the regions discussed.

P14L353-354: "the atmospheric moisture is mostly coming from the western Sahel region itself, in addition to some oceanic sources located in the central eastern Atlantic Ocean." Here and generally, I would find it helpful if the fractions of moisture from the different source regions could be quantified, e.g., through tables that list which percentage stems from the continental source region, which percentage from the oceanic source region, and which percentage from elsewhere. Regarding the western Sahel specifically, such numbers could be compared with numbers reported in Goessling and Reick (2013), according to which "only" 40% of the precipitation in the western Sahel stems from the African continent (consistently between different tracing methods, see Table 2 therein). Is it possible that this discrepancy is due to the short "lifetimes" used for the backward tracing, which leads to an overestimation of nearby contributions (as argued above)?

P15L363-365 and: "the interannual variability of RH in the region is strongly controlled by changes in the total annual precipitation and the total annual land evapotranspiration in the continental source region." - As detailed above, I think that the causal links are not sufficiently evidenced.

P15L374-375: "These relationships together would explain the observed trend in RH" - I find this paragraph confusing. In particular, the connection between correlations of interannual variations on the one hand and long-term trends on the other hand is not clear.

P15L378-379: "These results would suggest that RH has mostly changed over the West Sahel region, as a consequence of changes in the continental humidity sources" - Again, I think that the causal links are not sufficiently evidenced.

P15L380: "the same results" -> "the corresponding results"

P16L396-397: "Given the high control of these variables on the interannual variability of RH" - see my above comments on the causality issue.

P17L417-418: "other regions showed a weak correlation between the temporal variability of RH and land evapotranspiration in the moisture source region. A representative example is China" - I am wondering whether this might be explained largely by the fact that relative interannual ET variations are just much weaker in China (around 10% of the mean value) compared to other regions (20-30% of the mean value) so that the signal-to-noise ratio is worse in China?

P18L438-439: "Figure 10 depicts the relationship between RH and land evapotranspiration seasonally and annually at the global scale" - these are local ("pixel-by-pixel") correlations again, right? I recommend to clarify this, and in what way the interpretation differs from the previous analysis where RH in target regions is correlated with ET in corresponding source regions.

P19L474: "Island" -> "Continent"

P20L492-494: "although some regions showed positive changes in the oceanic evaporation, the amount of increase was much lower than that found for SST, suggesting a general positive trend in most of the world ÌĄs oceans" - I appears that this is consistent with the finding that, in contrast to q, "the global-mean precipitation or evaporation, commonly referred to as the strength of the hydrological cycle, does not scale with Clausius–Clapeyron" (Held and Soden 2006, in particular Fig. 2 therein).

Sect.4: In my view, a lot of the material presented here belongs rather into the introduction (e.g., L539-564; L575-584; L609-616; L629-644). I think that the main conclusions

from this paper could be worked out much more clearly if the references to other work were repeated here in a much more condensed form, so that they can be better linked with the authors' conclusions.

P22L534-536: "This finding highlights the importance of land evapotranspiration processes in defining RH variability over large world areas." - The causality again ...

P22L545-547: "Numerous model-based studies have supported the strong influence of land evaporation processes on air humidity and precipitation over land surfaces" - a very strong link has also been shown in Goessling and Reick (2011).

P23L565-567: "results indicate that humidity in the analyzed regions is largely originated over continental rather than oceanic areas." - I'd like to repeat (i) my suggestion to report percentages telling how much of the moisture arriving in the target regions stems from the different sources, and (ii) that the method used here might overestimate continental contributions.

P25L522: "since" - this word does not seem to make sense here; I suggest to rephrase the sentence.

P25L624-632: I fully agree that these other factors, in particular circulation variability and trends, introduce considerable uncertainties into analyses such as the one undertaken here.

Fig.9: It is not entirely clear to me in how far the "inter-regional" correlations shown here provide a different angle on the matter compared to the "intra-regional" correlations of interannual variations. Could the authors comment?

**References**

Goessling, H.F. and Reick, C.H., 2011. What do moisture recycling estimates tell us? Exploring the extreme case of non-evaporating continents. Hydrology and Earth System Sciences, 15, pp.3217-3235. doi:10.5194/hess-15-3217-2011

Goessling, H.F. and Reick, C.H., 2013. On the" well-mixed" assumption and numerical 2-D tracing of atmospheric moisture. Atmospheric Chemistry and Physics, 13, pp.5567-5585. doi:10.5194/acp-13-5567-2013

Held, I.M. and Soden, B.J., 2006. Robust responses of the hydrological cycle to global warming. Journal of Climate, 19(21), pp.5686-5699. doi:10.1175/JCLI3990.1
* * *

---

## Author Comment (AC1) · 16 Oct 2017

Anonymous Referee #1 The paper analyses changes in surface relative humidity (RH) in the last 35 years. It is mostly based on ERA-Interim reanalysis data, but observations (HadISDH) are also considered. The main technique used is a Lagrangian analysis of the moisture source regions. I think the topic is very important, and in scope for Earth System Dynamics. The paper will be a valuable contribution to the field, but in my opinion it requires some revisions.

We really appreciate the careful reading, the comments raised by the reviewer#1 and the positive assessment of the manuscript. Please find below the answers to each

[Figure]

comment and if you have any further concerns, please feel free to raise these new comments.

Major Issues 1. My main criticism is that the paper lacks clear conclusions. Section "Discussion and Conclusions" is dominated by a discussion of other relevant studies, with too little effort to distill out what is new. My suggestion is to separate out "Conclusions" to a separate section. There, clearly state what is new, what are the specific conclusions of this study.

We have included a separate section of Conclusions in the revised manuscript:

"The main conclusions of this study are: • There are dominant negative trends of RH and this decrease is mostly linked to the temporal evolution of RH during the boreal warm season. Negative trends do not show homogeneous spatial patterns, and some regions also show positive trends. • There is a high agreement between RH and specific humidity trends at the global scale, suggesting a moisture deficit in large areas to explain RH trends in opposition to atmospheric warming. • In general we found significant correlations between the interannual variability of land evapotranspiration and RH. • There are not correlation between the ratio of the air temperature over the target regions and SST in the source regions and the RH variability. • There is not a significant relationship between the interannual variability of the oceanic evaporation in the oceanic humidity source regions and RH in the target areas.

Given strong relevance of understanding current RH trends at the global scale, further research is still needed to consider other dynamic and radiative factors that may affect the temporal variability and trends of RH over continental regions."

2. l139-l151: Text and formulas for RH computation:

In this study we followed the formulation used by Willett et al. (2014) for the HadISDH RH dataset. The reason for this is to make better comparable the RH obtained from observations in the HadISDH and the RH obtained from the ERA-Interim dataset. This

has been stated in the revised manuscript.

a. Why is the decision, whether the ice or liquid equilibrium pressure is used, based on the wet bulb temperature (and not, for example, the physical temperature)? Please justify or change.

We took this decision following Willett et al. (2014):

"Where the calculated Tw values are below 0 ◦C, values of e are recalculated with respect to ice. This assumes that the wet bulb was in fact an ice bulb at that time and that the measurement was taken with a wet bulb thermometer. This potentially introduces a dry bias in q and e when T is near 0 ◦C. For RH, dry biases could be up to 4 % RH, increasing as Tw rises towards 0 ◦C."

In any case, we do not think this may have a key role in the interannual variability and trends of RH. The relationship found among HadISDH and ERA-Interim RH at the annual and seasonal scales is very strong and consistent spatially.

b. There are two different equilibrium pressures to consider, one for liquid water and one for ice. Authors chose to use the ice one for temperatures below 0◦C. There is no direct physical reason for this, since there will be large open water areas even at sub-zero air temperature and vice versa. But I guess it is a reasonable first choice if information on the surface itself is not available. However, the authors should be clear about that it is a choice they are making, and not dictated from physics. c. Generally, I find text and forulmas here confusing and outdated. From Eq. 1 it follows simply that RH = 100*e/es(T) = 100*es(Td)/es(T) where RH = relative humidity in percent, e = water vapor partial pressure, T = physical temperature, Td = dewpoint temperature, es() = equilibrium water vapor pressure. So, to calculate RH from Td, all that is needed is a valid parameterisation of es(T). An accepted modern one is given in Murphy, D. M. and T. Koop (2005), Review of the vapour pressures of ice and supercooled water for atmospheric applications, Q. J. R. Meteorol. Soc., 131(608), doi:10.1256/qj.04.94. (They give two different es(T) parameterisations, one for liquid, and one for ice.)

We really appreciate this suggestion. We will consider suggested parametrization for future studies. In this study, as stated above, we decided to follow the exact methodology followed by Willett et al. (2014) to facilitate the comparability between HadISDH (i.e. observed) and ERA-Interim RH (i.e. reanalyzed) datasets.

3. I would like to see an overview figure of the RH mean state and the E-P mean state, before the trends are discussed. Perhaps in the same style as Figure 1, for cold season, warm season, and anual mean. And also a short discussion, along with the figure. This is important to put the changes into perspective. Also, in my understanding the "null hypothesis", based on simple thermodynamic arguments and the simplistic assumption that the circulation does not change, is that the existing E-P pattern is enhanced under global warming. (The "dry gets drier, wet gets wetter paradigm (Held, Isaac M. and Brian J. Soden (2006), Robust Responses of the Hydrological Cycle to Global Warming, J. Climate, 19(21), 5686-5699, doi:10.1175/JCLI3990.1.).)

In the revised manuscript we have included a new figure with the mean RH and the Vertically Integrated Moisture Flux divergence and a short discussion. We have added the maps of the divergence of the Vertically Integrated Moisture Flux (VIMF) using data from Era-Interim instead E-P because, in general terms, VIMF divergence may be used to estimate regions where the precipitation dominates (negative values) over the evaporation (positive values).

"Supplementary Figure 1 shows the average seasonal and annual RH and the Vertically Integrated Moisture Flux (VIMF), which can be used to estimate regions where the precipitation dominates (negative values) over the evaporation (positive values), from the ERA-Interim dataset. RH shows higher average values over equatorial regions, Southeast Asia and the North Eurasia region. The lower values are recorded over tropical regions, mainly in the North Hemisphere. Spatial differences between the cold and warm regions are very low. The annual pattern of the VIFM over continents shows that precipitation exceeds evaporation over the Intertropical Convergence Zone, Southeast Asia and the islands between Pacific and Indian Oceans (Maritime

continent), a great part of South America, Central America, Central Africa, and north-ward to 40°N in the Northern Hemisphere. Evaporation is higher than precipitation over the main area of Australia, the Pacific coast of North America, Northeast Brazil, areas around Mediterranean Sea, Eastern coast of Africa and southwest Asia. Seasonally, it is evident the poleward movement of the ITCZ during the hemispheric summer, and the change of the pattern over North America and Eurasian continent."

Minor Issues The paper contains some English errors and idiosyncrasies. Example: "On the contrary", used in several places, where I think the authors mean "on the other hand". I recommend a careful proof-reading, preferably by a native speaker.

A careful proof-reading was conducted by a native speaker to avoid English errors.

l47 "water holding capacity": Please replace by "equilibrium amount of water vapor". (Holding capacity is physically wrong, since the air does not "hold" the water vapor in any way. The CC equation describes the equilibrium pressure of water vapor with liquid water.)

Replaced

l49 "could increase": Please replace by "is expected to increase".

Replaced

l83 "there are unavailable studies": Rearrange: "studies...are unavailable"

Replaced

l103 "challengeable" –> "challenging"

Replaced

l239 "moisture support": Perhaps replace by "moisture supply"? (Meaning of support is not clear here.)

Replaced

l277: "positive (E-P) field": I'm confused. Isn't Figure 1 showing just the RH trends? How does the E-P field enter the figure?

This is a mistake that has been solved in the revised manuscript.

l364 "controlled by": I would say "correlated to". How do you know what controls what?

Replaced

l378 [RH has increased] "as a consequence of changes in the continental humidity sources": Why have they increased?

This is detailed in the revised manuscript:

"...given the positive trend in annual precipitation".

l407 "air temperature and SST ratio" –> "and air temperature to SST ratio"

Replaced

l492 "Thus, although some regions showed positive changes in the oceanic evaporation, the amount of increase was much lower than that found for SST, suggesting a general positive trend in most of the world's oceans (Supplementary Figure 48, Supplementary Table 1)." Confusing. Did you mean: "Thus, although some regions showed positive changes in the oceanic evaporation, the amount of increase was much lower than that found for SST, which suggests that SST changes do not drive evaporation changes (Supplementary Figure 48, Supplementary Table 1)."

Thanks for the suggestion; it matches much better with what is intended to show.

l591 "This finding indicates that while different model experiments fully supported the hypothesis that the different warming rates between oceanic and continental areas can explain the projected decrease in RH under climate change conditions, our results for 14 different regions in the world are contradictory, given that most of these regions exhibited a negative RH trend for 1979- 2014." What is the contradiction? The differential

land sea warming mechanism predicts a decrease of RH over land, and you find a negative RH trend. I'm not sure if this is a language issue, or if there is a fundamental point that I'm missing.

This has been replaced in the revised manuscript:

"...the different warming rates between oceanic and continental areas can explain the projected decrease in RH under climate change conditions, our results for 14 different regions in the world show a non-clear influence of the air temperature to SST ratio to explain the observed RH trends."

l638 "Hadley Cell" –> "Hadley Cell (HC)"

Replaced

Figure 2: Please add a vertical zero line, so that one can judge whether trends are positive of negative.

The Figure already contains a vertical zero line. Probably it is an effect of the screen visualization.

Figures 6-8: Subplots are too small, titles almost impossible to read for me.

This is an issue for the large number of subplots included in the images but our intention is that these images appear in a full page so we will completely sure that titles are readable in the published manuscript.

Figures 10-11: "Signification" –> "Significance"

Replaced

Finally, we would like to thank the reviewer#1 for his/her effort on reviewing our manuscript and the good inputs suggested to improve it.
* * *
[Figure]

Suppl. Figure 1. Annual and seasonal average RH and Vertically Integrated Moisture
Flux (VIMF) from ERA-Interim dataset

ndly version

ion paper

---

## Author Comment (AC2) · 16 Oct 2017

H. F. Goessling (Referee)

**Summary ### The authors analyse changes in relative humidity (RH) over continental regions during the period 1979-2014. For a number of regions with clear positive or negative trends in RH, they relate the RH changes to possible trends in local air temperature and local precipitation as well as continental evapotranspiration (ET), SST, and ocean evaporation. The authors determine relevant source regions for which to compute the latter three quantities based on an existing Lagrangian vapour tracking algorithm. They analyse relationships between these quantities and RH not only with**

respect to trends, but also with respect to interannual variations. Generally, the clearest relations they find are positive correlations of RH with precipitation and ET, and they conclude that continental ET plays an important role as driver of continental RH changes.

The results shown in the paper are interesting and well presented (although the text needs quite a number of corrections in terms of language/grammar), and I think they merit publication. However, I am not convinced that the causality put forward in the interpretation is sufficiently evidenced, so I would recommend to phrase some conclusions more cautiously. Specifically, I would argue that the positive correlation between RH and continental ET does not prove the suggested causality. There are a number of other points, of which some are minor but some are not so minor, detailed below, that deserve clarification and/or revision. Overall, I recommend the manuscript should be reconsidered after major revisions.

We really appreciate the careful reading, the number of comments raised by Dr. Goessling and the general positive assessment of the manuscript. Please find below the answers to each comment and if you have any further concerns, please feel free to raise these new comments.

**Specific comments ### P1L20-22: "The aim was to account for the possible role of changes in air temperature over land, in comparison to sea surface temperature (SST), on RH variability." - This sentence seems to suggest that this was the only aim of the paper, but the role of land and ocean evapo(transpi)ration changes is obvisouly also accounted for ...**

This has been addressed in the revised manuscript as follows:

"The aim was to account for the possible role of changes in air temperature over land, in comparison to sea surface temperature (SST), but also the role of land evapotranspiration and the ocean evaporation on RH variability."

[Figure]

P1L22-25: "Results demonstrate a strong agreement between the interannual variability of RH and the interannual variability of precipitation and land evapotranspiration in regions with continentally-originated humidity." - After having read the paper, I wonder if the last part of this sentence is supported by the content: It appears that the authors have not systematically assessed how the fraction of "continentally-originated humidity" is related to the strength of this agreement. However, this would be quite interesting. I suggest to compute the fraction of continentally-originated humidity (sometimes called the continental recycling ratio) for each of the regions (and season), and to relate this fraction to the strength of the agreement to verify (or not) this statement.

We have addressed the suggested analysis in the revised manuscript. In the methodology section we have included an explanation about this:

"Also from FLEXPART simulations, we obtained the fractions of moisture from the continental and oceanic sources annually and for each cold and warm season. The purpose was to compare with the results obtained on the role of the land evapotranspiration and ocean evaporation of RH variability and trends."

In the results section (section 3.3) we have included the analysis suggested:

"In any case, attributing causality to the observed RH changes is quite complex given divergences found at the global scale. We have computed the fraction of continentally-originated humidity for each region and season and related this fraction to the strength of the agreement between RH and Land evapotranspiration at the annual and seasonal scales. Supplementary Table 1 shows the percentage of contribution of continental areas to the total moisture in each one of the fourteen analyzed regions, which oscillate between 31.6% for West Europe and 64% in Northeast Asia. There is not a significant relationship between these percentages of contribution and the strength of the agreement between RH and land evapotranspiration obtained in each region (Supplementary Figure 47). This reinforces the complexity of attributing changes of RH to a single factor. In any case, in some of the regions that show significant changes in RH

have been identified, there are also changes in the total contribution from continental areas at the seasonal and annual scales (Supplementary Figures 48-50). Both West Sahel and East Sahel show increased contribution of continental areas. On the contrary, La Plata region, in which there is also a strong agreement between RH and land evapotranspiration and that shows a significant negative trend in both variables, there is a decrease of the continental contribution. This stresses the complexity of giving a unique attribution to the observed RH changes."

P3L83: "Nonetheless, there are unavailable empirical studies that support ..." - In my view it would be more logical (and elegant) to rephrase this to something like the following: "However, there are no empirical studies available that support ...". (It seems that the term "nonetheless" is used in the sense of "however", which I think is not correct, also elsewhere in the paper.)

Replaced here and throughout the entire manuscript.

P3L84-86: "One of these hypotheses is related to the slower warming of oceans in comparison to continental areas. In particular, specific humidity of air advected from oceans to continents increases more slowly than saturation specific humidity over land. This would decrease RH over continental areas [...]. The observed decrease in RH over some coastal areas, which are adjacent to their sources of moisture, adds further uncertainty to this hypothesis." - Is the observed decrease in RH, be it over coastal areas or further inland, not rather supporting the hypothesis?

The cited studies show several findings of declines in RH over coastal areas. We do not state that these results invalidate the hypothesis but they add reasonable uncertainties as stated in the manuscript. Thus, under warmer conditions in regions close to water bodies it could be reasonably assumed a stationary RH instead a general RH decrease.

P4L87-96: As explained here, the land-atmosphere feedback seems to be "only" a positive feedback rather than one that could explain why RH is altered under global warming in the first place, no? We fully agree with your comment, and the term "global

warming" has been removed from the sentence.

P4L88-96 and P4L100-103: In this context (continental recycling ratios, evaporation-precipitation feedback, and the role of circulation changes), I cannot resist to recommend that our paper on exactly these aspects (Goessling and Reick 2011), where we have combined moisture tracing with an idealised perturbation experiment in a climate model, revealing that circulation changes play a major role, should also be considered; see also other related comments below.

Yes, really a perfect citation for this statement. Thanks.

P4L107: "advections that were" -> "advection that was"

Replaced

P5L125-128: Could the authors clarify how exactly the gap-filling was done?

We have detailed this in the revised manuscript:

"In order to avoid biases in the filling due to differences in the distribution parameters (mean and variance) between the candidate and the objective data series, a bias correction was performed on the candidate data. Thus, normal distribution was used for bias correction of RH. The data of the candidate series were re-scaled to match the statistical distribution of the observed series to be filled, based on the overlapping period between them."

Sect2.1: I suggest to use consistent headers for subsubsections 2.1.1-2.1.5, that is, either referring always to the dataset described in that section or to the variable(s).

Modified in the revised manuscript.

P6Eq1-8: Most of the equations need some corrections with respect to units. Note thatT and P have units which implies that some of the constants used need to have the same units.
We have included the units in the revised manuscript.

Sect2.1.5: While in the other sections the time period is always explicitly mentioned, that's not the case here; please clarify.

Time periods were addressed in the revised manuscript.

P8L190: "The statistical significance of the time series was tested at the 95% confidence interval" - I suggest it should be "trend" instead of "time series", and "level" instead of "interval".

Replaced in the revised manuscript.

P8L195: "with uneven number of stations" - I suggest this should be something like "with low station density".

Replaced in the revised manuscript

P9L213-235: First, it would be good to clarify in what way the particles are distributed in the vertical initially: Does their vertical distribution correspond to the specific humidity profile?

Of course, the vertical distribution is coincident with the specific humidity profile in the way that the ERA-Interim reproduces it. The FLEXPART model uses the whole levels of ERA-Interim to compute the specific humidity and the particles over a specific area take the value of q at the corresponding level. The model also ensures the existence of particles at all levels.

Second, I am wondering what explains the relatively short "optimal lifetimes" of 4-7 days found (as reported later in Sect. 3.2), and in particular what this implies. Given that even with the global-mean residence time of âĹij10 days only the closest (1-1/e)*100% of a (typical) source region is captured, it appears that only half or even less is captured on average with shorter backward tracking times. It appears that contributions from adjacent sources (in particular from nearby land) are thereby overestimated

compared to more remote contributions (in particular ocean). I suggest to clarify this.

We are aware of the great discussion that in recent times exists about the residence time of water vapor in the atmosphere. Depending on the approach you use to estimate it the average residence time can vary from 3-5 to 8-10 days or even more (see Läderach, A. and Sodemann, 2016 and its discussion supporting for shorter periods or van der Ent and Tuinenburtg, 2017 supporting for longer periods).

It has been usual to consider 10 days (as average), and this is the time used in most of the studies to compute moisture transport (including most of the previous studies by the authors). 10 days is not a magical number in the sense that there is a great variability depending on the season and the latitude considered. So in this work we preferred to compute an "optimal" time more than a real residence time, approach already used in our recent studies (e.g. Miralles et al. 2016). To do this we computed for each analyzed region "the most adjusted time" comparing the moisture transport for precipitation with precipitation data taken from a reanalysis. So first, the sources of moisture for each target region where calculated in a backward mode using 15 days of transport. Once the sources regions where delimited, we calculated in a forward mode the balance of E-P again during 15 days (we move all the particles departing each source and reaching the target region). Then we checked over the target region which was the "most adjusted time" of Flexpart result for E-P<0 ("precipitation") for each grid point, and finally we calculated the mean value of this "optimal adjusted time" for the whole target area. This is the time that we used in this study.

An explanation of this approach is included in the new version of the manuscript

Läderach, A. and Sodemann, H.: A revised picture of the atmospheric moisture residence time, Geophys. Res. Lett., 43, 924– 933, doi:10.1002/2015GL067449, 2016.

D.G. Miralles, R. Nieto, N.G. McDowell, W.A. Dorigo, N.E.C Verhoest, Y.Y. Liu, A.J. Teuling, A.J. Dolman, S.P. Good, L. Gimeno (2016) Contribution of water-limited ecoregions to their own supply of rainfall, Environmental Research Letters, vol 11, doi:

doi:10.1088/1748-9326/11/12/124007

van der Ent, R. J. and Tuinenburg, O. A.: The residence time of water in the atmosphere revisited, Hydrol. Earth Syst. Sci., 21, 779-790, https://doi.org/10.5194/hess-21-779-2017, 2017.

P9L226: "as better as" - bad grammar.

Replaced in the revised manuscript.

P9L232-233: "the optimal lifetime selected for each region was that fulfills the minimum absolute difference between" - bad grammar.

Replaced by:

"iii) the optimal lifetime selected for each region was chosen according to the minimum absolute difference between the FLEXPART simulated precipitation. . ."

P10L253-254: "the ratio between air temperature in the target region and SST in the source region" - If I am not mistaken, this is a quantity that depends on the units used for temperature (Kelvin or Degrees Centigrade). Also, wouldn't it be more straightworward to use simply the temperature difference instead of the ratio?

We have used the same units (°C). We think it is not relevant if the ratio or the difference is used to assess long term trends in the evolution of SST and land temperature.

P10L260: "signification" - Should be "significance", right? (Occurs many times throughout the manuscript.)

Replaced here and throughout the entire manuscript.

P11L263-267: "While a pixel-to-pixel comparison does not produce a reliable assessment of the possible contribution of land evapotranspiration to RH changes, given that the source of moisture can apparently be far from the target region, we still believe that this association can give insights on the global influence of land evapotranspiration on

RH changes." - Here and generally, I have the impression that the suggested causality is not sufficiently attested and discussed. I would argue that increased land ET tends to be caused primarily by increased precipitation (except in very humid regions), and that anomalous precipitation (caused, e.g., by circulation anomalies) is simply accompanied by corresponding RH anomalies. And I think this is still largely valid for a non-local comparison, where land ET is determined for the "source region", because (i) the source region tends to overlap strongly with the target region and (ii) also most of the non-overlapping part is rather close, where spatial correlations of synoptic-scale anomalies are still high. I suggest this could be the main explanation for the positive correlations between RH, precipitation, and land ET.

This has been included in the discussion section.

P12L290: "uneven distribution" -> "low density"?

Replaced

P12L302: "West Sahel" - Should be "East Sahel", right?

West Sahel is correct here. East Sahel shows a clear RH decrease.

P13L315-317: "On the contrary, areas of complex topography in the Northern Hemisphere, Australia, India, Northern South America and Africa showed positive trends." - Can the authors comment why this should be so?

This is analyzed in depth in section 3.2

P13L324-326: "high consistency between the HadISDH and the ERA-Interim datasets in terms of both the magnitude and sign of change of in RH (Supplementary Figures 2 and 3)" - Are these figures not showing the correlation of interannual variations instead of the "magnitude and sign of change" (where the latter sounds as if the long-term trend is referred to)?

This is true. The sentence has been rewritten as follows:

"Given this high consistency between the HadISDH and the ERA-Interim datasets in terms of both the magnitude and sign of change in RH (Figures 1 and 2) and also in interannual variations (Supplementary Figures 2 and 3),..."

P13L328-P14L337: If I am not mistaken, according to Figure 4, q has decreased less than it would have had to decrease in order to maintain RH constant (apparently due to a cooling trend?) in the mentioned regions (Southwest North America, the Amazonian region, Southern South America, and the (eastern!) Sahel). It appears that this corresponds to an INCREASE in RH in those regions, instead of the decrease shown in Fig. 1. Please clarify this contradiction.

No, it means that in these regions q has dominantly decreased (left), so to maintain a RH constant in these areas, according to the observed warming rate, there is a deficit of absolute humidity quantified in more than 2 g/kg-1. Negative value (in red) represent a deficit of moisture and positive values (in blue) represent an increase of q higher than that necessary to maintain Rh constant according to warming rates.

Sect.3.2: I suggest to structure this subsection with subsubsections corresponding to the regions discussed.

We followed this suggestion in the revised manuscript.

P14L353-354: "the atmospheric moisture is mostly coming from the western Sahel region itself, in addition to some oceanic sources located in the central eastern Atlantic Ocean." Here and generally, I would find it helpful if the fractions of moisture from the different source regions could be quantified, e.g., through tables that list which percentage stems from the continental source region, which percentage from the oceanic source region, and which percentage from elsewhere. Regarding the western Sahel specifically, such numbers could be compared with numbers reported in Goessling and Reick (2013), according to which "only" 40% of the precipitation in the western Sahel stems from the African continent (consistently between different tracing methods, see Table 2 therein). Is it possible that this discrepancy is due to the short "lifetimes" used

for the backward tracing, which leads to an overestimation of nearby contributions (as argued above)?

See above. This has been addressed in depth in the revised manuscript.

P15L363-365 and: "the interannual variability of RH in the region is strongly controlled by changes in the total annual precipitation and the total annual land evapotranspiration in the continental source region." - As detailed above, I think that the causal links are not sufficiently evidenced.

The sentence has been rewritten to avoid attributing causal links:

"As illustrated, the interannual variability of RH in the region is correlated to changes in the total annual precipitation and the total annual land evapotranspiration in the continental source region."

P15L374-375: "These relationships together would explain the observed trend in RH" - I find this paragraph confusing. In particular, the connection between correlations of interannual variations on the one hand and long-term trends on the other hand is not clear.

The sentence has been removed and the paragraph simplified.

P15L378-379: "These results would suggest that RH has mostly changed over the West Sahel region, as a consequence of changes in the continental humidity sources" - Again, I think that the causal links are not sufficiently evidenced.

Sentence has been removed.

P15L380: "the same results" -> "the corresponding results"

Replaced

P16L396-397: "Given the high control of these variables on the interannual variability of RH" - see my above comments on the causality issue.

The sentence has been removed in the revised manuscript.

P17L417-418: "other regions showed a weak correlation between the temporal variability of RH and land evapotranspiration in the moisture source region. A representative example is China" - I am wondering whether this might be explained largely by the fact that relative interannual ET variations are just much weaker in China (around 10% of the mean value) compared to other regions (20-30% of the mean value) so that the signal-to-noise ratio is worse in China?

It could be a possible explanation. In any case, we have included it as possible hypothesis for this pattern.

P18L438-439: "Figure 10 depicts the relationship between RH and land evapotranspiration seasonally and annually at the global scale" - these are local ("pixel-by-pixel") correlations again, right? I recommend to clarify this, and in what way the interpretation differs from the previous analysis where RH in target regions is correlated with ET in corresponding source regions.

This issue has been stressed in the revised manuscript.

P19L474: "Island" -> "Continent"

Replaced

P20L492-494: "although some regions showed positive changes in the oceanic evaporation, the amount of increase was much lower than that found for SST, suggesting a general positive trend in most of the world Ì ËŻAs oceans" - I appears that this is consistent with the finding that, in contrast to q, "the global-mean precipitation or evaporation, commonly referred to as the strength of the hydrological cycle, does not scale with Clausius–Clapeyron" (Held and Soden 2006, in particular Fig. 2 therein).

Yes, it is consistent and stated in the revised manuscript.

Sect.4: In my view, a lot of the material presented here belongs rather into the introduc-

tion (e.g., L539-564; L575-584; L609-616; L629-644). I think that the main conclusions from this paper could be worked out much more clearly if the references to other work were repeated here in a much more condensed form, so that they can be better linked with the authors' conclusions.

Following the suggestions raised by reviewer #1, the revised manuscript includes a new section of "Conclusion" to highlight the main findings of the research.

P22L534-536: "This finding highlights the importance of land evapotranspiration processes in defining RH variability over large world areas." - The causality again ...

Removed statement in the revised manuscript.

P22L545-547: "Numerous model-based studies have supported the strong influence of land evaporation processes on air humidity and precipitation over land surfaces" - a very strong link has also been shown in Goessling and Reick (2011).

Cited in the revised manuscript.

P23L565-567: "results indicate that humidity in the analyzed regions is largely originated over continental rather than oceanic areas." - I'd like to repeat (i) my suggestion to report percentages telling how much of the moisture arriving in the target regions stems from the different sources, and (ii) that the method used here might overestimate continental contributions.

See above. This has been addressed in depth in the revised manuscript. In any case, although the FLEXPART methodology could overestimate continental contribution as the reviewer suggests, we would like to stress that major analysis are not based on the total moisture provided by different sources obtained by means of the FLEXPART scheme but using the GLEAM data set, which is independent. FLEXPART was mostly used to identify the surface area corresponding to the main oceanic and continental moisture sources and we consider that the possible surface overestimation or underestimation has a minor impact here.

P25L522: "since" - this word does not seem to make sense here; I suggest to rephrase the sentence. Sentence has been rewritten.

P25L624-632: I fully agree that these other factors, in particular circulation variability and trends, introduce considerable uncertainties into analyses such as the one undertaken here.

Thanks.

Fig.9: It is not entirely clear to me in how far the "inter-regional" correlations shown here provide a different angle on the matter compared to the "intra-regional" correlations of interannual variations. Could the authors comment?

Really this figure does not provide a different angle in comparison to the analysis in the specific regions but we think that it is relevant to summarize the findings obtained in the different regions to provide if there is a possible spatial relationship.

**References ### Goessling, H.F. and Reick, C.H., 2011. What do moisture recycling estimates tell us? Exploring the extreme case of non-evaporating continents. Hydrology and Earth System Sciences, 15, pp.3217-3235. doi:10.5194/hess-15-3217-2011**

Goessling, H.F. and Reick, C.H., 2013. On the" well-mixed" assumption and numerical 2-D tracing of atmospheric moisture. Atmospheric Chemistry and Physics, 13, pp.5567-5585. doi:10.5194/acp-13-5567-2013

Held, I.M. and Soden, B.J., 2006. Robust responses of the hydrological cycle to global warming. Journal of Climate, 19(21), pp.5686-5699. doi:10.1175/JCLI3990.1

Finally, we would like to thank Dr. Goessling for his effort on reviewing our manuscript and the constructive inputs suggested to improve it.

| Region | Cold season | Warm season | Annual |
|---|---|---|---|
| West Europe (region 1) | 18.3 | 45.1 | 31.6 |
| Scandinavia (region 2) | 17.5 | 46.5 | 33.1 |
| Central-East Europe (region 3) | 24.5 | 56.2 | 41.1 |
| South-East Europe and Turkey (region 4) | 26.2 | 66.5 | 49.2 |
| West Sahel (region 5) | 48.6 | 58.1 | 54.0 |
| India (region 6) | 56.5 | 48.8 | 51.8 |
| East China (region 7) | 51.9 | 54.2 | 53.3 |
| North East Asia (region 8) | 36.7 | 76.4 | 64.0 |
| La Plata (region 9) | 38.7 | 52.1 | 45.9 |
| Canada (region 10) | 20.1 | 66.3 | 47.4 |
| Central USA (region 11) | 28.4 | 64.6 | 49.3 |
| West North America (region 12) | 20.4 | 52.6 | 36.7 |
| Amazonian (region 13) | 31.3 | 32.0 | 31.7 |
| East Sahel (region 14) | 54.7 | 66.1 | 61.0 |

Supplementary Table 1: Percentage of moisture coming from the continental source in
each one of the fourteen analyzed regions obtained from the FLEXPART model

**Fig. 1.**

[Figure]

[Figure]

Suppl. Figure 47: Relationship between the average fraction of moisture coming from continental areas and the strength of the agreement between RH and Land Evapotranspiration obtained in each region (by means of the Pearson's r coefficient between RH and Land Evapotranspiration).

**Fig. 2.**

[Figure]

Suppl. Figure 48: Evolution of the percentage of continentally-originated humidity for each of the fourteen regions at the annual scale. The magnitude of change (in %) and statistical significance of the trend is indicated.

**Fig. 3.**

Suppl. Figure 49: As in Suppl. Figure 48, but during the cold season.

Fig. 4.

[Figure]

Suppl. Figure 49: As in Suppl. Figure 48, but during the warm season.

**Fig. 5.**

---

## Author Response (AR2)

Dear Dr. Kleidon:

Please find attached a revision of the manuscript entitled "Recent changes of relative humidity: regional connection with land and ocean processes" to be considered for publication in Earth System Dynamics. In the revised manuscript, we have addressed all comments and suggestions raised by you and the third reviewer. You will also find enclosed a letter that includes a detailed point to point response to all comments.

We look forward to hearing from you at your earliest convenience, and should you have any questions please feel free to contact us.

Sincerely,

Sergio M. Vicente-Serrano and coauthors

**Editor Comments:**

**Your manuscript was re-reviewed by a third reviewer. This reviewer raised some concerns similar to one of the reviewers of the first round, namely that the causality of your results in some parts of the manuscript are not sufficiently attested and discussed. This reviewer recommended rejection as the reviewer considered the current version far from being acceptable for publication, but encouraged to resubmit after these issues are addressed. The concern of the reviewer particularly refers to section 3.3. I agree with the reviewers assessment that this section takes an approach that is too simple to interpret RH changes in terms of evaporation changes. Correlation does not imply causation, especially when dealing with the hydrologic cycle with its very tight linkages between evaporation, humidity, and precipitation. Also note that the aerodynamic term in evaporation is typically quite small, and evaporation is predominantly limited by energy. So it is unclear to me what the correlation between RH and evaporation is supposed to imply.**

We understand the concerns raised by the reviewer. We agree with the reviewer that in the earlier version we lacked the opportunity to provide a detailed and comprehensive interpretation and discussion of the obtained results. Moreover, in our attempt to attribute the possible physical mechanisms driving the observed RH, we probably biased the interpretation of the results toward a higher importance of land evapotranspiration processes, given the obtained empirical relationships. In the revised manuscript, we have carefully considered this by making a re-elaboration of the interpretation of the obtained results, and entirely rewriting the discussion and conclusion sections.

In the revised manuscript, we agree that RH correlation with several physical mechanisms does not imply true causality. Within the text, we have not established this direct possible causation related to the evapotranspiration variability and trends. Correspondingly, we also stress the cases in which a coherent and reasonable connection between RH and land evapotranspiration can be attributed given not only on the correlation among the two variables but also the coherence of the detected trends and the contribution of continental and oceanic sources to moisture supply in specific regions. This is widely discussed below in our response we provide to the reviewer.

In relation to the comments related to the aerodynamic and radiative connections, we agree that the radiative component can be important in explaining the magnitude of the Atmospheric Evaporative Demand (AED). Here, I would like to establish a distinction between the AED, which is exclusively driven by the atmospheric variables that control the aerodynamic (air temperature, Relative humidity and wind speed) and radiative (downward direct and diffuse radiation) components and the real (or actual) evapotranspiration (ET) (better than evaporation, since transpiration is much more important than direct evaporation), which does not only depend on the AED but also on the soil water availability. In arid and semiarid regions, land evapotranspiration is

mostly driven by soil water availability and thus the AED exceeds very much ET. In humid climates, where there are no water constrains, ET is mostly constrained by the AED.

There is evidence that the role of the radiative component on the average magnitude of the AED (here different forms of AED could be valid for this statement, e.g. Pan evaporation, reference evaporation, ETo) may be strongly variable at the global scale and in some regions the aerodynamic component is even more important in explaining the magnitude of the AED than the radiative component. In any case, and irrespective of the relative importance of both components on the AED magnitude, the existing evidences on this issue not only at the global scale, but also in regional studies, suggest that the aerodynamic component explains most of the observed temporal variability of the AED. Thus, it is important not only to consider the sensitivity of the AED to the different meteorological variables, but also the observed trends of the variables that control the AED. Observations suggest that meteorological variables that control the aerodynamic component show-on average- more changes than the incoming solar radiation (e.g. McVicar et al., 2012a; J. Hydrol., and 2012b J.; Ecohydrology) and the aerodynamic component is the main explanatory factor of the recorded AED trends at the global scale (Wang et al., 2012, J. Clim., 25, 8353–8361). We also analysed this issue in depth in Spain (Vicente-Serrano et al., 2014 Water Resources Research 50, 8458–8480) using observational 50-yrs data, concluding that AED is more sensitive to temperature and relative humidity than to solar radiation. We found that the strong AED trends recorded over the last 50 years are mostly driven by the strong decrease in RH. A similar pattern is found also in the Canary Islands (Vicente-Serrano et al., 2016, Hyd. Earth Sys. Sci. 20, 3393-3410). Other studies have stressed the role of wind speed in particular regions (e.g. Gu et al., 2018: Atmosphere 9: 9; Wang et al. 2017: J.Hydrol 544: 97).

Nevertheless, and irrespective of the comments raised in this revision on the importance of the meteorological drivers of the AED, we have not analyzed the possible role of aerodynamic and radiative components on land evapotranspiration, as well as the possible influence of land evapotranspiration, oceanic evaporation and moisture transport issues on the temporal variability and trends of RH. Although this issue is of particular relevance, given the strong influence of RH on the AED, this issue is out of the scope of this research, particularly with the uncertainty in attributing RH to different physical processes. Alternatively, in this first-time comprehensive empirical retrospective analysis, we have stressed the ways in which both oceanic and land mechanisms can probably contribute to explaining the spatially complex RH trends found in this study. We agree with the reviewer that oceanic contribution is essential in large regions of the world and some mechanisms related to the different warming between land and oceanic areas could contribute to better explanation and understanding of the decrease of RH in some regions. Nevertheless, land contribution is also important in explaining RH and AED anomalies  in some regions. This has been

stressed in a number of studies, as we have indicated in the discussion section, as follows:

*"...In the same context, there is strong evidence that low levels of soil moisture and land evapotranspiration are usually accompanied by a reinforcement of low RH, particularly during drought episodes. Under these circumstances, the suppression of the latent heat flows from the soil to the atmosphere would enhance soil and vegetation warming and sensible heat, inducing air temperature rise. Also, the lack of supply of water vapor to the atmosphere favors the decrease of RH and the reinforcement of severity of heat waves (Hirschi et al., 2011). Seneviratne et al. (2002) showed that vegetation control on transpiration might contribute significantly to enhancement of summer drying, particularly when soil water is limited. Other studies confirmed this finding for other regions worldwide, employing both observational data (e.g. Hisrchi et al., 2011) and model outputs (e.g. Seneviratne et al., 2006; Fischer et al., 2007). Our study suggests good spatial agreement between changes in RH and those of continental contribution to precipitation as well as land evapotranspiration during summertime. Although this finding is markedly evident for all the analyzed regions, it should be seen with caution. This is mainly because physical processes driven soil moisture are more active during the warm season (Vautard et al., 2007 and 2013; Miralles et al., 2014), which adds difficulty to establish full causality between RH and other driving forces during this season."*

This is a relevant and desired research topic to analyze in depth (see e.g. a recent granted ERC project that focusses on the role of land evapotranspiration on AED and how these issues may cause aggravation and/or spatial propagation of droughts given the decrease in air moisture and land supply anomalies; http://www.dry2dry.org/context).

**Furthermore, at the moment, Section 5 merely summarizes the findings, but they are not conclusions. I think there is more to say from your results, so more work should be done on formulating what the analysis actually implies regarding the causes for RH trends.**

We have rewritten the conclusions section of the manuscript, including an assessment of the findings related to the possible drivers of the observed RH trends:

[revised manuscript text omitted]

**Comments by reviewer 3:**

**This study investigated the change of RH and its relationship with land and ocean hydrological variables. My major comment is the same as that by an earlier reviewer (H. F. Goessling): "Here and generally, I have the impression that the suggested causality is not sufficiently attested and discussed. I would argue that increased land ET tends to be caused primarily by increased precipitation (except in very humid regions), and that anomalous precipitation (caused, e.g., by circulation anomalies) is simply accompanied by corresponding RH anomalies. I suggest this could be the main explanation for the positive correlations between RH, precipitation, and land ET."**

**About the contribution of nonlocal moisture sources, the strong correlation also does not suggest causality, because all the variables in the hydrological cycle could be controlled by atmospheric circulation, leading to strong correlation. Please refer to the following paper, and it can provide you some clue on the relationships among RH, SST and ocean ET.**

**Wei, J., Q. Jin, Z.-L. Yang, P. A. Dirmeyer, 2016: Role of ocean evaporation in California droughts and floods, Geophysical Research Letters, 43, 6554–6562, doi: 10.1002/2016GL069386.**

**In summary, although the study investigated an important and interesting topic, it did not produce any major finding that is new and useful and did not give any solid conclusion on the causal relationships. Therefore, the paper is still far from acceptable for publication.**

Here, our intention was not to provide a causality of the temporal variability and trends in RH at the global scale. Alternatively, we have stressed the strong complexity of this issue. As opposed to earlier studies that have employed modelling approaches with GCM scenarios to determine the possible connection between RH trends and different physical mechanism related to moisture transport from oceanic areas, the differences between the warming of oceanic and continental areas (since they affect air saturation), but also the role of land evaporation processes. In this study, we have employed empirical information, reanalysis data and statistical tools to establish possible relationships that may give insights on the possible drivers of RH temporal variability and trends.

A detailed response to this point is outlined in our response to Dr. H.F. Goessling, which was included in the revised manuscript. As we completely agree that correlation does not necessarily imply causality, we have been careful with the issue of "causality", This aspect has been stressed several times in the revised manuscript:

*"In any case, attributing causality to the observed RH changes is quite complex given divergences found at the global scale. We have computed the fraction of continentally-originated humidity for each region and season and related this fraction to the strength*

*of the agreement between RH and Land evapotranspiration at the annual and seasonal scales. Supplementary Table 1 shows the percentage of contribution of continental areas to the total moisture in each one of the fourteen analyzed regions, which oscillate between 31.6% for West Europe and 64% in Northeast Asia. There is not a significant relationship between these percentages of contribution and the strength of the agreement between RH and land evapotranspiration obtained in each region (Supplementary Figure 47). **This reinforces the complexity of attributing changes of RH to a single factor.** In any case, in some of the regions that show significant changes in RH have been identified, there are also changes in the total contribution from continental areas at the seasonal and annual scales (Supplementary Figures 48-50). Both West Sahel and East Sahel show increased contribution of continental areas. On the contrary, La Plata region, in which there is also a strong agreement between RH and land evapotranspiration and that shows a significant negative trend in both variables, there is a decrease of the continental contribution. **This stresses the complexity of giving a unique attribution to the observed RH changes."***

We completely agree with the reviewer that moisture supply from oceanic regions can contribute significantly to explaining precipitation (and RH) variability over large continental regions. As such, the discussion section was entirely rewritten to clarify this kind of issues and avoid any misinterpretation of what can be really inferred with the statistical analysis of different empirical information applied here.

Nevertheless, we would like also to stress that the scope of our study was not to explain the general contribution of continental and/or oceanic sources to RH. Instead, we try to understand the mechanisms that drive the occurrence of different trends in RH at the global scale, in connection to trends in land evapotranspiration, oceanic evaporation, and (now in the revised manuscript) with the moisture supply from oceanic and continental sources.

In the same context, we would like also to stress that anomalies in land evapotranspiration, accompanied with low soil moisture, may cause changes in RH, given the balance between latent and sensible fluxes. We have stressed this point in the revised manuscript, with no direct causality using our empirical information:

*"...In the same context, there is strong evidence that low levels of soil moisture and land evapotranspiration are usually accompanied by a reinforcement of low RH, particularly during drought episodes. Under these circumstances, the suppression of the latent heat flows from the soil to the atmosphere would enhance soil and vegetation warming and sensible heat, inducing air temperature rise. Also, the lack of supply of water vapor to the atmosphere favors the decrease of RH and the reinforcement of severity of heat waves (Hirschi et al., 2011). Seneviratne et al. (2002) showed that vegetation control on transpiration might contribute significantly to enhancement of summer drying, particularly when soil water is limited. Other studies confirmed this finding for other regions worldwide, employing both observational data (e.g. Hisrchi et al., 2011) and model outputs (e.g. Seneviratne et al., 2006; Fischer et al., 2007). Our*

*study suggests good spatial agreement between changes in RH and those of continental contribution to precipitation as well as land evapotranspiration during summertime. Although this finding is markedly evident for all the analyzed regions, it should be seen with caution. This is mainly because physical processes driven soil moisture are more active during the warm season (Vautard et al., 2007 and 2013; Miralles et al., 2014), which adds difficulty to establish full causality between RH and other driving forces during this season.”*

We agree with the reviewer that atmospheric circulation is an essential factor for better understanding of climate variability, so dynamic processes must be taken into account to provide an overall picture of the variability and trends of RH at the global scale. Thus, we clearly stressed this issue in the discussion section of the revised manuscript, as follows:

*“Sherwood (1996) indicated that RH distributions are strongly controlled by dynamical fields rather than local air temperatures. This suggests that atmospheric circulation processes could largely affect the temporal variability and trends of RH. A range of studies indicates noticeable changes in RH, in response to low-frequency atmospheric oscillations, such as the Atlantic Multidecadal Oscillation (AMO) and El Niño-Southern Oscillation (e.g. McCarthy and Toumi, 2004; Zhang et al., 2013), regional circulation (Wei et al., 2016a and 2016b), as well as changes in the Hadley Cell (HC) (Hu and Fu, 2007). Wright et al. (2010) employed a global climate model under double $CO_2$ concentrations to show that tropical and subtropical RH is largely dependent on a poleward expansion of the Hadley cell: a deepening of the height of convective detrainment, a poleward shift of the extratropical jets, and an increase in the height of the tropopause. Also, Laua and Kim (2015) assessed changes in the HC under $CO_2$ warming from the Coupled Model Intercomparison Project Phase-5 (CMIP5 model projections. They suggest that strengthening of the HC induces atmospheric moisture divergence and reduces tropospheric RH in the tropics and subtropics. This spatial pattern resembles the main areas showing negative trends in RH in Northern as well as Southern hemisphere.”*

We have revised the discussion section, following the comments raised by the reviewer and relevant discussion in other related articles. Again, our objective was not to connect variability and changes of RH with atmospheric circulation processes, but to relate with temporal variability and trend of some climate/hydrological/oceanic variables that can drive the temporal variability and the trends of RH. As we stressed above, we have not tried to establish causality, but rather to stress the complexity of the mechanisms and to explore statistically significant relation that may provide some clues on the factors affecting RH variability and trends worldwide. We believe that modelling approaches are valuables tools to unravel the complex physical mechanisms behind RH variability and trends. Studies based on empirical information are also essential to evaluate the confidence of the suggested processes and the statistical robustness of the expected coherent relationships.

**Some specific comments:**

**1. Instead of land ET and ocean ET, can you calculated the relationship of RH with the land and ocean moisture contribution to precipitation (E-P)? I guess the RH relationship with ocean moisture contribution should be stronger than with ocean ET.**

In the revised manuscript, we analyzed the relationships between RH and land and oceanic moisture contribution to precipitation (E-P). In the revised manuscript, we have included a new section (3.2), in which we explain in depth these relationships.

**2. L78. add "increases" after "specific humidity"**

Added

**3. L247. I think 95th percentile of E-P is not a good criterion because there are very large areas have very small E-P values. It is better to use 95% moisture supply criterion, i.e., the selected area contributes 95% of the moisture for the target region's precipitation.**

We have selected the 95th percentile to limit the higher extension area for the sources of moisture, to account for the main sources, and to plot a continuous line in the figures. In previous works, we have adopted the same threshold or high (98th percentile) to achieve a better and realistic contribution of moisture. In any way, the percentile selected is near the 95% of the values, due the fact that the field has a continuous pattern in its extension, without jumps that could affect the calculation of percentage or/and percentiles.

**4. Page 12. From line 290. The discussion for supplementary Figure 1 is very long. If it worth such long discussion, it should be a formal figure, not a supplementary figure.**

Supplementary Figure 1 has been included in the main manuscript, as Figure 1.

**5. L290-291. Vertically Integrated Moisture Flux (VIMF). How is it calculated? Is it E-P from FLEXPART? I feel it is not the normal VIMF (qv) people talking about because in Line 300 you mentioned "Evaporation is higher than precipitation over …"**

VIMF is calculated directly from the ERA-Interim reanalysis data. Evaporation and precipitation in this interpretation are related with divergence and convergence field in the VIMF plot.

**6. L296 "regions" should be "seasons"?**

Replaced

**7. Figure 2. You should add some figures that shows only the ERA-Interim data with the same available grids as the HadISDH. In this way, the comparison is more accurate.**

We have included the suggested figure, as Supplementary Figure 1.

[Figure]

Supplementary Figure 1: Spatial distribution of the magnitude of change of RH (% per decade) over the period 1979-2014 from HadISDH (left) and ERA-Interim dataset (right) considering the points with available HadISDH observatories

**8. L730. "not" should be "no"**

Replaced.

---

## Author Response (AR3)

Dear Dr. Kleidon:

Please find attached a revision of the manuscript entitled "Recent changes of relative humidity: regional connection with land and ocean processes" to be considered for publication in Earth System Dynamics. In the revised manuscript, we have addressed all minor comments raised by the third reviewer; the existing typos have been removed, and the length of the manuscript has been noticeably reduced. We have removed 8 pages of text and three figures regarding the previous version of the manuscript.

We look forward to hearing from you at your earliest convenience, and should you have any questions please feel free to contact us.

Sincerely,

Sergio M. Vicente-Serrano and coauthors

[revised manuscript text omitted]
 andstatistically significant (positive trends at $p < 0.05$) trends (blue), positive statistically insignificant positive trends (cyan), negative statistically insignificant negative trends (orange) and negative andstatistically significant negative trends (red).

[Figure]

Figure 4: Scatterplots showing the global relationship between the magnitude of change in RH with HadISDH stations and ERA-Interim dataset at the seasonal and annual scales. Colors represent the density of points, with red color showing the highest density of points.

[Figure]

Figure 5Fig. 3: Spatial distribution of the seasonal and annual magnitudes of change in specific humidity (g/kg$^{-1}$) (left) and the deficit/surplus of specific humidity to maintain the RH constant withat the levels of 1979 according to the land air temperature evolution (from the CRU TS v.3.23 dataset) for 1979-2014.

[Figure]

Figure 6: Distribution of the selected 14 world regions, based on the high consistency in RH trends between the HadISDH and the ERA-Interim datasets. These regions were selected for the identification of the oceanic and land humidity sources by means of the FLEXPART scheme.

[Figure]

Figure 7Fig. 5: Top left: Annual RH humidity trends in the WestWestern Sahel (region 6), Top right: average (E-P)>0 at the annual scale to identify the main humidity sources in the region (mm. year$^{-1}$). Center: RelationshipRelationships between the de-trended annual RH and the de-trended annual variables for 1979-2014. Bottom: Annual evolution of the different variables corresponding to the WestWestern Sahel region. The magnitude of change and their corresponding statistical significance of the trend isare indicated for each variable.

[Figure]

Figure 8Fig. 6: The same as Fig. 75, but for the La Plata (region 9).

[Figure]

Figure 9Fig. 7: The same as Fig. 75, but for the West Northof Northern America (region 12).

[Figure]

Figure 10: RelationshipFig. 8: Relationships between the annual oceanoceanic contribution to annual precipitation (E-P) and the annual RH in the target regions.

[Figure]

Figure 11: RelationshipFig. 9: Relationships between the annual land contribution to annual precipitation (E-P) and the annual RH in the target regions.

[Figure]

Figure 12Fig. 10: Evolution of the land contribution (%) to annual precipitation (%) in the different target regions

[Figure]

Figure 13: RelationshipFig. 11: Relationships between the average annual magnitudemagnitudes of change in RH identified infor each one of the 14 analyzed regions and the annual magnitudemagnitudes of change in precipitation, the ratio between air temperature/SST, oceanic evaporation and land evapotranspiration.

[Figure]

Figure 14: Spatial distribution of the Pearson's r correlations between the detrended RH and land evapotranspiration series at the annual and seasonal time scales. The statistical significance of the correlations is also shown.

[Figure]

Figure 15Fig. 12: Spatial distribution of the magnitude of change in the annual and seasonal land evapotranspiration (1979-2014) and their corresponding statistical significance of trends.

[Figure]

Figure 16Fig. 13: Annual and seasonal magnitude of change of SST and OAFLUX oceanic evaporation for 1979-2014.